# Myogenic regulatory transcription factors regulate growth in rhabdomyosarcoma

Inês M Tenente[1,2,3], Madeline N Hayes[1,2], Myron S Ignatius[1,2,4], Karin McCarthy[1,2], Marielle Yohe[5], Sivasish Sindiri[5], Berkley Gryder[5], Mariana L Oliveira[1,2,6], Ashwin Ramakrishnan[1,2], Qin Tang[1,2], Eleanor Y Chen[7], G Petur Nielsen[8], Javed Khan[5], David M Langenau[1,2]*

[1]Molecular Pathology, Cancer Center, and Regenerative Medicine, Massachusetts General Hospital, Boston, United States; [2]Harvard Stem Cell Institute, Cambridge, United States; [3]GABBA Program, Abel Salazar Biomedical Sciences Institute, University of Porto, Porto, Portugal; [4]Molecular Medicine, Greehey Children's Cancer Research Institute, San Antonio, United States; [5]Oncogenomics Section, Pediatric Oncology Branch, Advanced Technology Center, National Cancer Institute, Gaithersburg, United States; [6]Instituto de Medicina Molecular, Faculdade de Medicina, Universidade de Lisboa, Lisbon, Portugal; [7]Department of Pathology, University of Washington, Seattle, United States; [8]Department of Pathology, Massachusetts General Hospital, Boston, United States

*For correspondence:
dlangenau@mgh.harvard.edu

Competing interests: The authors declare that no competing interests exist.

**Abstract** Rhabdomyosarcoma (RMS) is a pediatric malignancy of muscle with myogenic regulatory transcription factors MYOD and MYF5 being expressed in this disease. Consensus in the field has been that expression of these factors likely reflects the target cell of transformation rather than being required for continued tumor growth. Here, we used a transgenic zebrafish model to show that Myf5 is sufficient to confer tumor-propagating potential to RMS cells and caused tumors to initiate earlier and have higher penetrance. Analysis of human RMS revealed that MYF5 and MYOD are mutually-exclusively expressed and each is required for sustained tumor growth. ChIP-seq and mechanistic studies in human RMS uncovered that MYF5 and MYOD bind common DNA regulatory elements to alter transcription of genes that regulate muscle development and cell cycle progression. Our data support unappreciated and dominant oncogenic roles for MYF5 and MYOD convergence on common transcriptional targets to regulate human RMS growth.

## Introduction

Continued tumor growth and relapse are driven by molecularly defined tumor propagating cells (TPCs). These TPCs share molecular and functional properties with non-transformed stem cells. For example, molecularly defined TPCs can divide to produce daughter cells with similar functional characteristics as the parental TPC, akin to the process of self-renewal found in normal stem cells. TPCs can also divide to produce mature, specialized cells that have specific functions within the developing cancer. Given that TPCs drive continued tumor growth, it is not surprising that these cells are often retained following treatment and ultimately drive refractory, metastatic, and relapse disease (*Reya et al., 2001*). TPCs have been identified in a wide range of cancers, including both zebrafish and human rhabdomyosarcoma (RMS), a devastating pediatric cancer of muscle (*Langenau et al., 2007*; *Chen et al., 2014*; *Ignatius et al., 2012*). Yet, to date, the molecular mechanisms driving TPC frequency and sustained tumor growth in RMS have not been fully defined. Moreover, it is unknown the extent to which normal muscle stem cell programs play a role in regulating RMS growth.

Rhabdomyosarcoma is a common sarcoma characterized by impaired muscle differentiation. These tumors express myogenic regulatory transcription factors (MRFs) including Myogenic factor 5 (MYF5) and Myoblast determination protein D (MYOD) (*Clark et al., 1991*; *Kumar et al., 2000*; *Parham, 2001*; *Sebire and Malone, 2003*) and are used in the clinical diagnosis of RMS. RMS is comprised of two molecular subtypes. Alveolar RMS (ARMS) harbor *Pax7-FOXO* and *Pax3-FOXO* genomic fusions (*Sorensen et al., 2002*) and have few additional recurrent genomic changes (*Chen et al., 2013b*; *Shern et al., 2014*). By contrast, 90% of human embryonal rhabdomyosarcoma (ERMS) have RAS pathway activation and a higher mutation burden when compared with ARMS (*Chen et al., 2013b*; *Langenau et al., 2007*; *Shern et al., 2014*). Common mutations found in ERMS include inactivation of *Tp53* and activating mutations of *FGFR4, PDGFA*, and *NOTCH1* (*Chen et al., 2013b*; *Shern et al., 2014*). Yet, roles for these pathways in regulating TPC number and proliferation have not been reported. In fact, to date, only the Sonic-Hedgehog and canonical WNT/B-catenin signaling pathways have been shown to regulate TPC function in a subset of human RMS (*Chen et al., 2014*; *Satheesha et al., 2016*). Understanding additional underlying mechanisms of TPC growth and function will be important for defining new therapies to treat pediatric RMS.

Despite the similarity of RMS cells with embryonic and regenerating muscle and well-known roles for the myogenic regulatory transcription factors MYF5 and MYOD in regulating these processes, their role in driving RMS growth has yet to be reported. Rather, it has been suggested that activation of the myogenic gene regulatory programs likely reflect the target cell of transformation and may not be required for continued RMS growth (*Keller and Guttridge, 2013*; *Kikuchi et al., 2011*; *Macquarrie et al., 2013b*; *Rubin et al., 2011*). Despite MYF5 and MYOD being highly expressed in human and animal models of RMS (*Langenau et al., 2007*; *Rubin et al., 2011*), exerting important roles in muscle development and stem cell self-renewal in regeneration (*Buckingham and Rigby, 2014*), and being able to reprogram fibroblasts into proliferating myoblasts (*Braun et al., 1989*; *Tapscott et al., 1988*); a functional requirement for these transcription factors in regulating RMS growth has gone unexplored since their discovery over two decades ago.

Transgenic zebrafish models have become a powerful tool to uncover new biological insights into human cancer (*Langenau et al., 2003, 2007*; *Le et al., 2007*; *Park et al., 2008*; *Patton et al., 2005*; *Sabaawy et al., 2006*; *Yang et al., 2004*; *Zhuravleva et al., 2008*). In the setting of ERMS, we have developed a mosaic transgenic zebrafish that express human $kRAS^{G12D}$ under control of the *rag2* minimal promoter, which is expressed in lymphoid cells (*Jessen et al., 2001*; *Langenau et al., 2003*) and muscle progenitor cells (*Langenau et al., 2007*). Thus, when $kRAS^{G12D}$ was expressed under control of this promoter, 20–40% mosaic injected fish developed ERMS (*Langenau et al., 2007*). Because 10–20 transgene copies are commonly integrated into the genome (*Langenau et al., 2008*), one can inject multiple transgenes into one-cell stage embryos with stable integration and expression being observed in developing tumors. Using this mosaic transgenic approach, we can deliver transgenic expression of $kRAS^{G12D}$, a fluorescent label to mark ERMS cells, and a modifying gene to assess synergies in regulating tumor initiation (*Langenau et al., 2008*). Importantly, the zebrafish model accurately mimics many of the molecular underpinnings of the human disease and has been used to uncover important genes and pathways relevant to human cancer (*Chen et al., 2013a, 2014*; *Ignatius et al., 2012*; *Langenau et al., 2007*; *Le et al., 2013*). The model has also been used to identify functional heterogeneity in molecularly defined cell types, including isolation of *myf5:GFP+* TPCs (*Ignatius et al., 2012*). In total, the zebrafish $kRAS^{G12D}$ ERMS model has emerged as one of the most relevant for discovering pathways that drive cancer growth in human RMS (*Chen et al., 2013a, 2014*; *Ignatius et al., 2012*; *Kashi et al., 2015*; *Langenau et al., 2007, 2008*; *Le et al., 2013*; *Storer et al., 2013*; *Tang et al., 2016*)

Here we show that *Myf5* is not only a marker of TPCs in the zebrafish ERMS model (*Ignatius et al., 2012*), but was sufficient to impart tumor propagating potential to differentiated ERMS cells in vivo. *Myf5* re-expression also lead to tumors that initiated earlier, had higher penetrance, and were larger than $kRAS^{G12D}$-alone expressing ERMS. Experiments in human RMS uncovered significant inter-tumoral heterogeneity of MRF expression with high MYF5 or MYOD defining largely mutually exclusive groups of tumors. Functional studies showed that both MYF5 and MYOD are required for continued RMS proliferation, likely acting redundantly with one another to regulate common molecular programs found in normal muscle development and regeneration. Consistent with this interpretation, ChIP-seq analysis identified common binding sites of MYF5 and MYOD in promoter and enhancer regions of genes that regulate cell cycle and muscle differentiation. A subset

of these same genes were confirmed to be downregulated upon MYF5 or MYOD knockdown. Finally, we show that MYF5 and MYOD are also required for efficient human RMS tumor growth in vivo. Our data supports a previously unappreciated role for MYF5 and MYOD in regulating growth, proliferation, and TPC activity in rhabdomyosarcoma.

## Results

### Re-expression of *myf5* in zebrafish ERMS cells accelerated tumor onset and increased penetrance

We have uncovered that *myf5* is highly expressed in undifferentiated, molecularly defined TPCs in zebrafish $kRAS^{G12D}$-induced ERMS (*Langenau et al., 2007*; *Ignatius et al., 2012*). Remarkably, this TPC fraction shares molecular and functional properties with non-transformed muscle satellite stem cells. For example, cell transplantation and direct live cell imaging has revealed that *myf5:GFP+/* myosin-negative progenitor cells drive tumor growth and specifically label TPCs in this animal model (*Ignatius et al., 2012*; *Chen et al., 2014*). To assess roles for *myf5* in regulating ERMS growth, we transgenically expressed *myf5* under control of the differentiated myosin light chain muscle pro-moter (*mylpfa*). This transgene faithfully drives expression in terminally-differentiated muscle cells in both transient and stable transgenic fish (*Xu et al., 1999*; *Langenau et al., 2007*; *Ignatius et al., 2012*; *Storer et al., 2013*; *Chen et al., 2014*) and has been used to identify zebrafish ERMS cell sub-fractions that lack *myf5*, have low proliferative capacity, cannot self-renew, and do not sustain ERMS growth in vivo (*Ignatius et al., 2012*). Here, $rag:kRAS^{G12D}$ was co-injected with *mylpfa:myf5* into one-cell-stage zebrafish and analyzed for tumor onset.

Histological analysis was performed on ERMS tumors arising in $rag2\text{-}kRAS^{G12D}$;*mylpfa-myf5* AB-strain transgenic fish and compared with those that express only $kRAS^{G12D}$ (*Figure 1A–F*, *Figure 1—figure supplement 1*). Tumors were histologically staged based on differentiation (*Storer et al., 2013*; *Figure 1—figure supplement 2*). As reported previously, primary $kRAS^{G12D}$-induced ERMS were comprised of 50% undifferentiated stage 1 ERMS (N = 5 of 10, *Figure 1B,C* and *Figure 1—figure supplement 2*), which harbored mostly small round blue cells. By contrast, *mylpfa:myf5* expressing primary ERMS contained only 7.7% stage 1 ERMS (N = 2 of 26, p=0.015, Chi-square test, *Figure 1E–F*), with the remaining tumors being highly differentiated stage 2 and 3 ERMS (*Figure 1F* and *Figure 1—figure supplement 2*). These tumors had large numbers of rhabdomyoblasts and cells with fibrous and spindle cell morphology. Transcriptional profiling of bulk tumor cells by qRT-PCR confirmed that *mylpfa:myf5* expressing ERMS cells had high *myf5* transgene expression, were more differentiated, and yet also had elevated expression of TPC-associated markers including *c-met* and *cadherin 15* (*Figure 1G*, *Langenau et al., 2007*; *Ignatius et al., 2012*). These gene markers are also commonly expressed in zebrafish muscle progenitor and satellite cells (*Siegel et al., 2013*; *Gurevich et al., 2016*). Collectively, these data show that re-expression of *myf5* in myosin-expressing ERMS cells leads to tumors with differentiated morphology and are consistent with the re-activation of muscle stem cell programs in differentiated cell types.

Tumors arising in double transgenic $rag2:kRAS^{G12D}$; *mylpfa:myf5* expressing ERMS were also larger by 30 days postfertilization than those that expressed only $kRAS^{G12D}$ (*Figure 1H*, p=0.0108, Student's t-test). However, apoptosis was not altered following re-expression of *myf5* (*Figure 1—figure supplement 3*). Proliferation was assessed by both phospho-histone H3 staining and EDU incorporation following intra-peritoneal injection and assessed at 6 hr. From this analysis, we uncovered wide variation in proliferation between tumors, with a trend toward increased proliferation in *mylpfa:myf5* expressing ERMS when assessed by EDU incorporation (*Figure 1—figure supplement 3D*). Together, our data support a model where *mylpfa:myf5* expressing ERMS initiate earlier and with higher penetrance than those that express only $kRAS^{G12D}$ (*Figure 1I*, p<0.001, log-rank Mantel-Cox test), likely reflecting a dominant role for transgenic *myf5* in transforming a wider range of cell types and to a lesser degree on elevating proliferation.

To confirm that differentiation changes were confined to fully transformed ERMS cells, we next assessed the histology of ERMS following transplantation into immune-deficient $rag2^{E450fs}$ recipient fish (*Figure 2A–F*; *Tang et al., 2014*). $kRAS^{G12D}$-expressing ERMS were comprised exclusively of undifferentiated stage one tumors (*Figure 2B,C* and *Figure 2—figure supplement 1*, n=10 trans-planted fish arising from four independent tumors). By contrast, ERMS that re-expressed *myf5* had

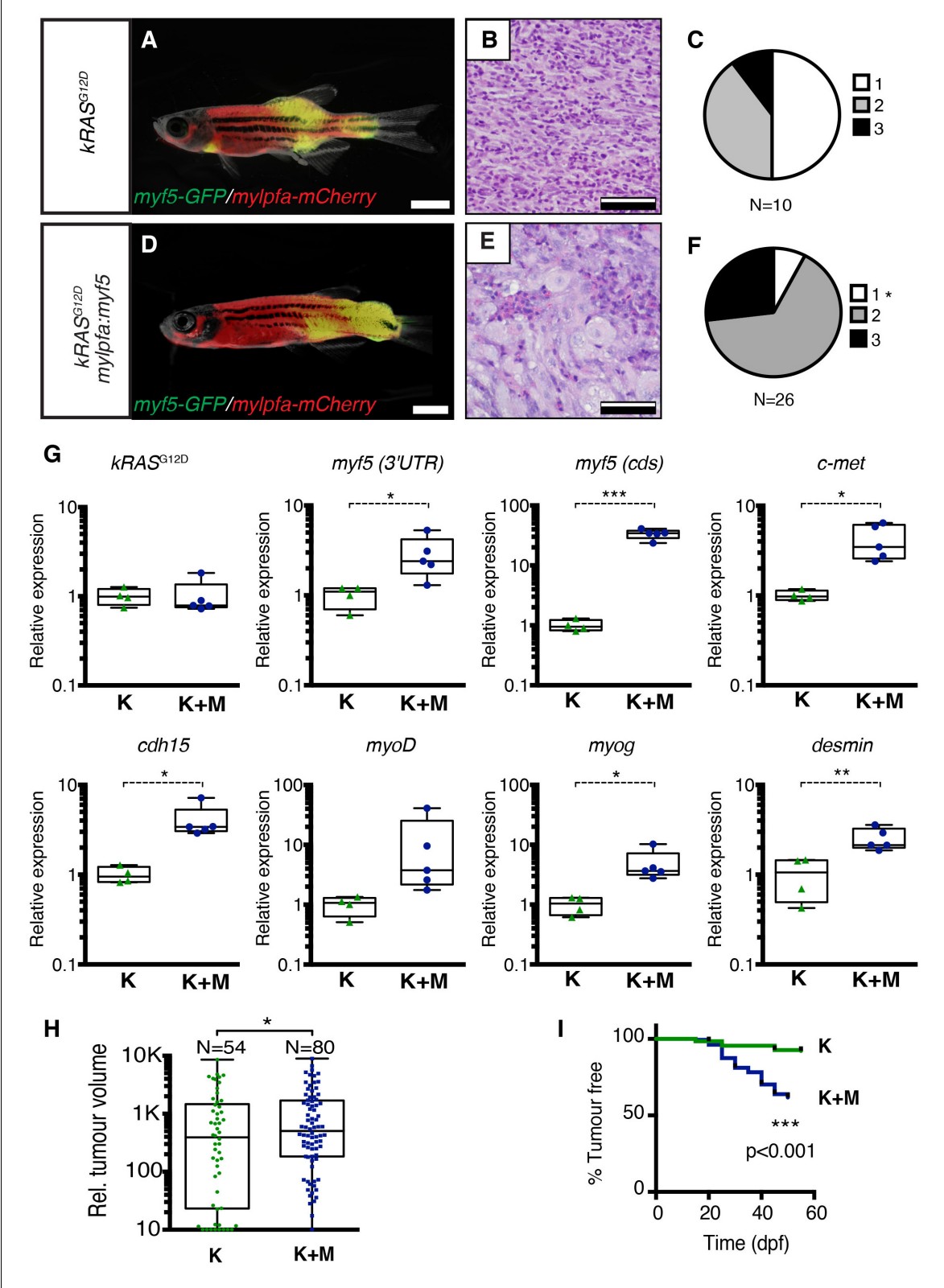

**Figure 1.** Transgenic *myf5* elevates tumor cell differentiation, increases tumor size, and accelerates time to primary tumor-onset when expressed in myosin-expressing ERMS cells. (A–F) Primary ERMS developing in *myf5:GFP/mylpfa:mCherry* AB-strain zebrafish. Transgenic *kRAS^G12D*-expressing ERMS (A–C) compared with those that express both *kRAS^G12D* and *mylpfa:myf5* (D–F). Animals imaged at 35 dpf (A,D). Hematoxylin and Eosin-stained sections of representative tumors (B,E) and quantification of differentiation within individual tumors (C,F; 1-less differentiated and 3-most differentiated).

*Figure 1 continued on next page*

*Figure 1 continued*

Asterisk denotes p=0.015 by Chi-square test. (G) Quantitative real-time PCR gene expression performed on bulk ERMS cells, confirming high *myf5* expression, increased differentiation, and high expression of TPC associated genes in ERMS that co-express *kRAS^G12D* and *mylpfa:myf5* (K+M, N = 5). Endogenous *myf5* was assessed using primers specific to the 3'UTR and total *myf5* assessed by primers that amplify the coding sequence (cds). *cadherin 15* (*cdh15*) and *myogenin* (*myog*). *kRAS^G12D* alone expressing ERMS (K, N = 4). Average gene expression with 50% confidence intervals denoted by box. Mean, maximum, and minimum also denoted. (H) Relative tumor size of primary ERMS at 30 days post fertilization (dpf). Box shows 50% confidence interval. Mean, maximum, and minimum denoted. Asterisk denotes p=0.0108, Student's t-test. (I) Kaplan-Meijer analysis denoting time-to-tumor onset (p<0.001, Log-rank Statistic, N = 494 fish analyzed for K and N = 470 for K+M). Scale bars equal 2 mm (A,D) and 50 μm (B,E). Asterisks in panels G-H denote *p<0.05; **p<0.01; ***p<0.001 by Student's t-test.

The following figure supplements are available for figure 1:

**Figure supplement 1.** Fluorescence images of primary ERMS developing in stable transgenic *myf5:GFP/mylpfa:mCherry* zebrafish.
**Figure supplement 2.** Histological classification of primary zebrafish ERMS based on differentiation score.
**Figure supplement 3.** Analysis of proliferation and apoptosis in zebrafish primary ERMS.

differentiated histology and were comprised exclusively of stage 2 and 3 tumors (*Figure 2E,F* and *Figure 2—figure supplement 1*, n=15 transplanted fish from four independent tumors, p<,0.001, Chi-square test). Consistent with our histological evaluation, flow cytometric analysis revealed that differentiated, *mylpfa:mCherry-positive* (R+) tumor cells were greatly expanded in ERMS that aberrantly express *myf5* (*Figure 2G–I*, p=0.006, Student's t-test). These same tumors had reduced numbers of *myf5:GFP* (G+) and double-positive (G+R+) cells. As was seen in primary ERMS, *mylpfa:myf5* expressing ERMS also initiated earlier and with higher penetrance when engrafted into *rag2^e450fs*-recipient animals (*Figure 2J*, 2.5 × 10^5 cells/animal, p=0.046, Mantel-Cox log-rank statistic). These tumors also had a trend toward being larger when assessed at 30 days post-transplantation (*Figure 2K*). Effects on ERMS differentiation were confirmed in transplanted CG1 strain syngeneic animals, showing that *mylpfa-myf5* expressing ERMS were more differentiated based on morphology (*Figure 3—figure supplement 1A–F*, p<0.01, Chi-square test) and contained larger numbers of differentiated, *mylpfa-mCherry-positive* (R+) ERMS cells (*Figure 2L*, p<0.001, Student's t-test). These transplanted tumors also had significant reductions in *myf5-GFP* (G+) and double-positive (G+R+) ERMS cells. Together, these data confirm that *mylpfa:myf5* expressing ERMS were fully transformed and exhibited a more differentiated cellular phenotype when compared with *kRAS^G12D* alone expressing ERMS (*Figure 2E,F* and *Figure 2—figure supplement 1*).

## *Myf5* reprograms differentiated ERMS cells into TPCs

Because endogenous *myf5* expression labels molecularly defined TPCs in zebrafish *kRAS^G12D*-induced ERMS (*Ignatius et al., 2012*; *Chen et al., 2014*), we next questioned if TPC frequency might be altered in *mylpfa:myf5* expressing ERMS. Specifically, *rag2:kRAS^G12D* was co-injected with or without *mylpfa:myf5* into one-cell stage, CG1 syngeneic *myf5:GFP/mylpfa:mCherry* transgenic animals (*Mizgireuv and Revskoy, 2006*; *Ignatius et al., 2012*). Following tumor growth in primary transplant recipients (*Figure 3A–C*), cell subpopulations were isolated by FACS and transplanted into secondary syngeneic recipient fish at limiting dilution (*Figure 3D–I*, *Figure 3—figure supplement 1G–P*, 1x10^3−10 cells/animal, purity >85%, and >95% viability). As previously reported (*Ignatius et al., 2012*), only the *myf5:GFP+* (G+) single-positive ERMS cells from *kRAS^G12D*-alone expressing ERMS could efficiently engraft tumors into CG1-strain syngeneic recipient animals (*Figure 3J* and *Table 1*; N = 3 independent tumors). By contrast, both the *myf5:GFP+* single-positive (G+) and differentiated *myf5:GFP+; mylpfa:mCherry+* double positive (G+R+) ERMS cells could engraft disease when isolated from *mylpfa:myf5* expressing ERMS (N = 3 tumors analyzed, *Table 1* and *Figure 3L*; p=0.0002, ELDA analysis). Importantly, engrafted tumors displayed similar differentiated histology following engraftment with sorted cells when compared with primary tumors (*Figure 3B,E,H*). Quantitative real-time PCR of sorted cell fractions also showed largely similar myogenic gene expression in ERMS cell subfractions isolated from either *kRAS^G12D*-expressing or *kRAS^G12D*+ *myf5* expressing ERMS, confirming that our cell lineage labeling approach identified

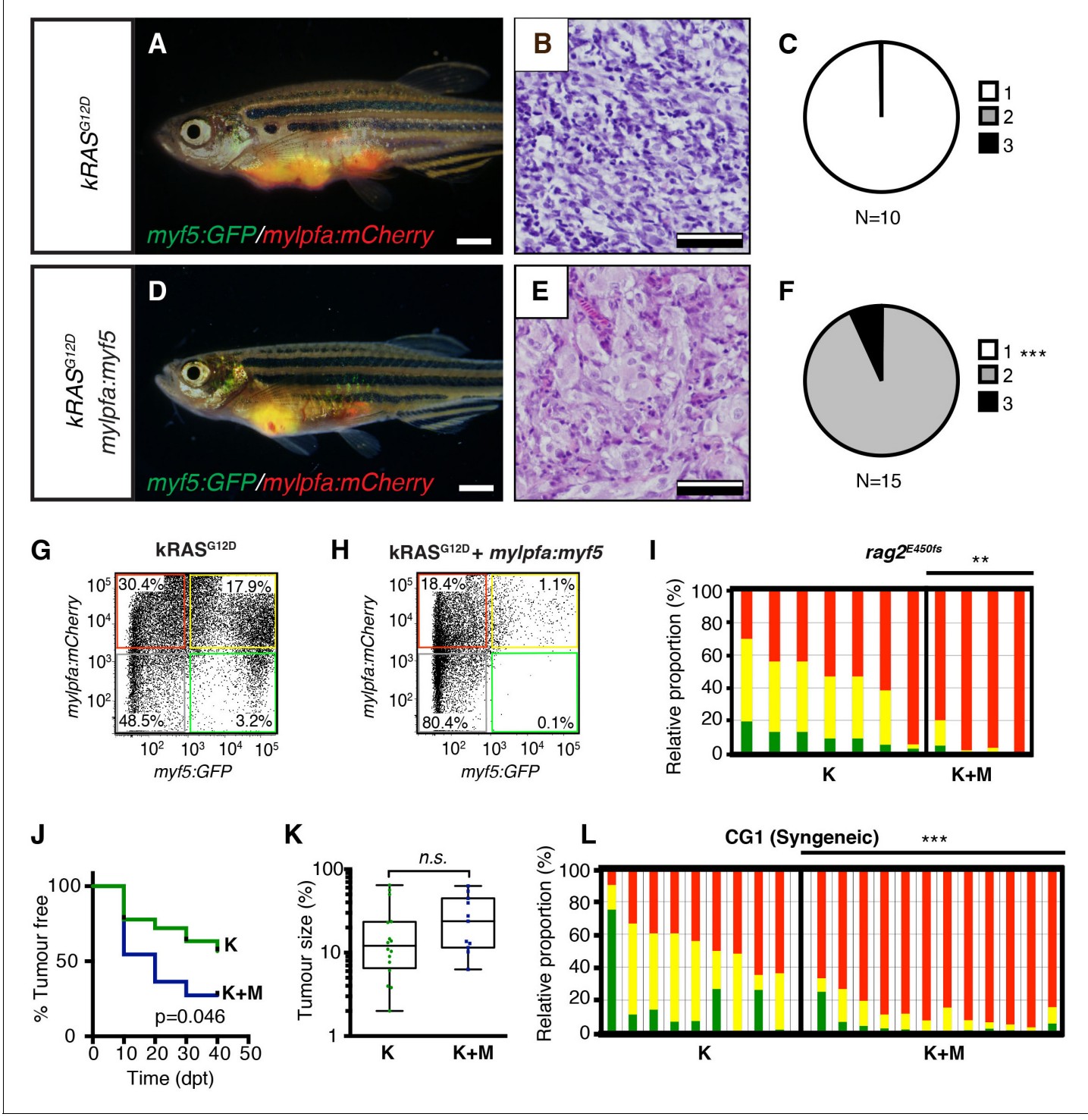

**Figure 2.** Tumors that transgenically express *myf5* are fully transformed and retain a differentiated phenotype following engraftment into recipient animals. (**A–F**) Analysis of ERMS arising in transplanted fish. *kRAS^{G12D}* expressing ERMS arising in *rag2^{E450fs}* transplant recipient fish (**A–C**) compared with those that express both *kRAS^{G12D}* and *mylpfa:myf5* (**D–F**). Tumors were created in stable transgenic *myf5:GFP/mylpfa:mCherry* transgenic, AB-strain zebrafish and imaged following engraftment into recipient fish at 30 days post transplantation (dpt). Hematoxylin and eosin stained sections of representative tumors (**B,E**) and quantification of differentiation within individual ERMS (**C,F**; 1-less differentiated and 3-most differentiated). Asterisks denote p<0.001 by Chi-square test. (**G,H**) Representative flow cytometry analysis of fluorescently-labeled ERMS cells isolated from transplanted *rag2^{E450fs}* zebrafish. (**I**) Graphical summary of ERMS cell sub-fractions that grow following engraftment into immune-deficient *rag2^{E450fs}* recipients. Individual tumors are represented as separate bars with the proportion of G+ (green), G+R+ (yellow) and R+ (red) sub-populations denoted. **p=0.006.
*Figure 2 continued on next page*

*Figure 2 continued*

(J) Kaplan-Meijer analysis showing time-to-tumor onset in transplanted ERMS arising in *rag2*$^{E450fs}$ zebrafish (p=0.046, Log-rank Statistic, $2 \times 10^5$ cells/fish, N > 12 animals per arm, representing ≥3 independently-arising primary ERMS). (K) Relative tumor size at 30 days post engraftment (same animals analyzed as in J). (L) ERMS cells were also more differentiated following engraftment of *myf5:GFP/mylpfa:mCherry* ERMS cells into syngeneic recipient fish (p<0.001, Student's T-test, N ≥ 3 independently arising primary ERMS and assessed in n ≥ 2 animals per transplanted tumor). Scale bars equal 2 mm (A,D) and 50 μm (B,E).

The following figure supplement is available for figure 2:

**Figure supplement 1.** Histological classification of transplanted zebrafish ERMS based on differentiation score.

similar molecularly-defined subpopulations of ERMS cells in these tumors. One notable exception was *myf5*, which was also highly expressed in the G+R+ population of *mylpfa:myf5* transgenic tumor as expected (*Figure 3K,M*). Taken together, these data show that re-expression of *myf5* can lead to acquisition of tumor propagating potential in differentiated *mylpfa*-expressing ERMS cells in the zebrafish model.

## MYF5 and MYOD are required for continued tumor growth in human RMS

To explore the role of *MRFs* in human RMS, we next assessed *MYF5* and *MYOD* transcript expression in human primary tumor samples. Analysis of microarray gene expression (N = 133 samples) (*Davicioni et al., 2009*) and RNA-sequencing (RNA-seq) data sets (N = 98 samples) (*Shern et al., 2014*) uncovered that *MYOD* and *MYF5* were expressed along with specific muscle genes and defined two distinct gene regulatory modules in human RMS. One gene module included the co-expression of *MYF5*, *MYF6* and *PAX7* while the other expressed *MYOD* and higher levels of *CDH15*, and *MYOG* (*Figure 4A,B*). This correlation in gene expression was seen in comparison of all human RMS (*Figure 4A,B*) or within specific RMS subtypes (*Figure 4—figure supplement 1*), suggesting that *MYF5* and *MYOD* likely sit atop a transcriptional hierarchy to regulate muscle-specific gene programs in RMS. We next assessed a panel of human RMS cell lines for expression of MYF5 and MYOD following Western blot analysis. Remarkably, we found that the expression of these proteins was largely mutually exclusive in human RMS cell lines (N = 7, *Figure 4C*), suggesting that these proteins may act redundantly to regulate human RMS growth. This analysis also uncovered that only the Rh18 ERMS cells expressed MYF5 in our panel of human cell lines. MYF5 and MYOD expression were also assessed at the single cell level through immunofluorescence and verified that MYF5 and MYOD were mutually exclusively expressed in Rh18 and RD cells (*Figure 4—figure supplement 2*). Collectively, our data show significant inter-tumoral heterogeneity in the expression of myogenic factors in human RMS and suggests convergence of these transcription factors on regulating a common set of genes that are likely required for RMS growth.

Human Rh18 ERMS cells express high levels of MYF5 and were utilized in loss-of-function studies to assess roles in regulating proliferation, growth, and apoptosis. MYF5 protein expression was effectively reduced following si-*MYF5* mediated knockdown (*Figure 4D*) and resulted in significant impairment of cell proliferation as assessed by EdU-incorporation and flow cytometry (*Figure 4E–F*, p<0.001, Student's t-test). For example, si-*MYF5* treated cells showed a remarkable 70% reduction in S-phase cycling cells following 48 hr of treatment (p<0.001, Student's t-test, *Figure 4E–F*). Apoptosis was not increased in *si-MYF5* treated cells at 72 hr but lead to increased numbers of apoptotic cells by 96 hr (*Figure 4—figure supplement 3B*). These results were independently confirmed using stable knockdown with three independent lentiviral shRNAs specific to *MYF5* (*Figure 4G–I*, protein knockdown ranged from 50–95%). All sh-*MYF5* knockdown cells showed a remarkable cell cycle arrest with a virtual abrogation of S-phase cycling cells (*Figure 4H–I* and *Figure 4—figure supplement 3A*; p<0.001, Student's t-test). This phenotype was not associated with an overall increase in apoptosis (*Figure 4—figure supplement 3C,D*) and yet lead to a significant 60% decrease in cell number as assessed by manual nuclei counts performed at 96 hr and compared with shRNA control treated cells (*Figure 4—figure supplement 3E*). We conclude that MYF5 loss results in impaired cell cycle and secondarily elevates apoptosis.

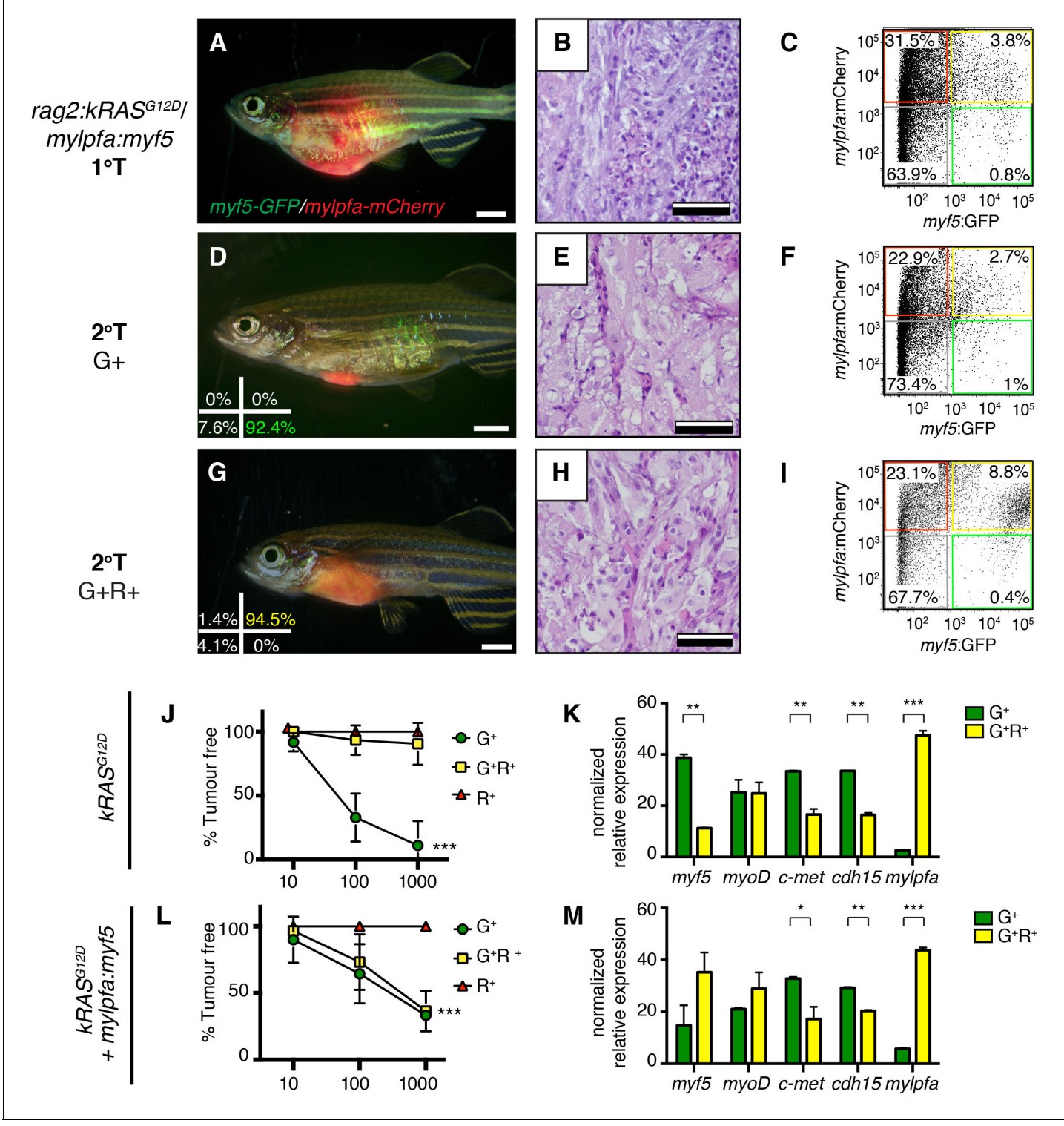

**Figure 3.** Limiting dilution cell transplantation shows that *myf5* can confer tumor-propagating ability to differentiated *myf5:GFP+/mylpfa:mCherry+* cells. Tumors were generated in *myf5:GFP/mylpfa:mCherry* CG1-strain syngeneic zebrafish. Representative tumors arising in primary transplanted fish (1°T, A–C) or secondary transplanted fish following engraftment with highly purified *myf5:GFP+, mylpfa:mCherry*-negative (2°T G+, D–F) or *myf5:GFP+, mylpfa:mCherry+* ERMS cells (2°T G+R+, G–I). Sort purity following FACS is noted in the lower left panels of D and G and was >92% for each population. These cells were used for cell transplantations and data provided in D-I. Cell viability was >95%. (J,L) Graphical summary of tumor engraftment following limiting dilution cell transplantation using highly purified sorted ERMS cells. Data is combined from all tumors shown in *Table 1*. ***p<0.0002 by ELDA analysis. (K,M) Relative gene expression analysis of sorted G+ or G+R+ ERMS cells from representative *kRAS^{G12D}* (K) or

*Figure 3 continued*

kRAS^{G12D}; mylpfa:myf5 (**M**) expressing ERMS (Standard Deviation, n = 3 technical replicates per PCR condition). *p<0.05; **p<0.01 and ***p<0.001 by Student's t-test.

The following figure supplement is available for figure 3:

**Figure supplement 1.** Analysis of transplanted ERMS arising in CG1-strain syngeneic recpients.

To assess if MYOD can also drive continued tumor growth and proliferation in human RMS, we next performed *MYOD* knockdown in human RD ERMS cells. siRNA knockdown resulted in reduced proliferation and a striking reduction in S-phase cycling cells (***Figure 4J–O***; p<0.001, Student's t-test). In keeping with our model that apoptotic cell death was induced secondary to cell cycle defects, we found that *si-MYOD* treatment had no effect on viability after 72 hr of knockdown, while analysis at 96 hr post-treatment resulted in elevated cell death of RD knockdown cells (***Figure 4— figure supplement 3G***). shRNA knockdown showed similar effects on suppressing cell cycle progression and growth (***Figure 4M–O*** and ***Figure 4—figure supplement 3F,H***). In fact, RD cell number

**Table 1.** *myf5* confers tumor-propagating ability to differentiated *myf5-GFP+/mylpfa-mCherry+* ERMS cells. Engrafted animals per cell dose are noted. Experiments for three independent tumors are shown. G+ (*myf5-GFP+/mylpfa-mCherry-*), G+R+ (*myf5-GFP+/mylpfa-mCherry+*), R+ (*myf5-GFP-/mylpfa-mCherry+*), DN (*myf5-GFP-/mylpfa-mCherry-*). Not applicable (NA); tumor-propagating cell frequency (TPC Freq.); 95% confidence interval (95% CI). Lower panel denotes cumulative TPC frequency for all three ERMS analyzed per genotype. Asterisk denotes p=0.0002 by ELDA analysis.

| kRAS^{G12D} Tumor #1 | | | | | kRAS^{G12D} + mylpfa:myf5 Tumor #1 | | | | |
|---|---|---|---|---|---|---|---|---|---|
| Cell # | G^+ | G^+R^+ | R^+ | DN | Cell # | G^+ | G^+R^+ | R^+ | DN |
| 1000 | 6 of 6 | 2 of 7 | 0 of 6 | 0 of 7 | 1000 | 2 of 3 | 4 of 5 | 0 of 6 | 0 of 6 |
| 100 | 5 of 9 | 0 of 9 | 0 of 8 | 0 of 10 | 100 | 6 of 10 | 2 of 10 | 0 of 8 | 0 of 7 |
| 10 | 0 of 8 | 0 of 8 | 0 of 9 | 0 of 7 | 10 | 3 of 10 | 1 of 10 | 0 of 10 | 0 of 8 |
| TPC Freq. | 1 in 140 | 1 in 3561 | NA | NA | TPC Freq. | 1 in 81 | 1 in 477 | NA | NA |
| 95% CI | 59–329 | 872–13740 | NA | NA | 95% CI | 40–165 | 201–1129 | NA | NA |
| kRAS^{G12D} Tumor #2 | | | | | kRAS^{G12D} + mylpfa:myf5 Tumor #2 | | | | |
| Cell # | G^+ | G^+R^+ | R^+ | DN | Cell # | G^+ | G^+R^+ | R^+ | DN |
| 1000 | 6 of 6 | 0 of 6 | 0 of 6 | 0 of 6 | 1000 | 2 of 3 | 1 of 2 | 0 of 7 | 0 of 7 |
| 100 | 4 of 7 | 2 of 10 | 0 of 10 | 0 of 10 | 100 | 1 of 6 | 3 of 6 | 0 of 7 | 0 of 10 |
| 10 | 1 of 8 | 0 of 9 | 0 of 10 | 0 of 8 | 10 | 0 of 8 | 0 of 9 | 0 of 10 | 0 of 8 |
| TPC Freq. | 1 in 109 | 1 in 3495 | NA | NA | TPC Freq. | 1 in 809 | 1 in 467 | NA | NA |
| 95% CI | 44–270 | 808–15120 | NA | NA | 95% CI | 244–2685 | 137–1589 | NA | NA |
| kRAS^{G12D} Tumor #3 | | | | | kRAS^{G12D} + mylpfa:myf5 Tumor #3 | | | | |
| Cell # | G^+ | G^+R^+ | R^+ | DN | Cell # | G^+ | G^+R^+ | R^+ | DN |
| 1000 | 2 of 3 | 0 of 2 | 0 of 3 | 0 of 4 | 1000 | 2 of 3 | 3 of 5 | 0 of 3 | 0 of 3 |
| 100 | 8 of 9 | 0 of 8 | 0 of 8 | 1 of 8 | 100 | 3 of 10 | 1 of 10 | 0 of 9 | 0 of 10 |
| 10 | 1 of 8 | 0 of 9 | 0 of 9 | 0 of 9 | 10 | 0 of 10 | 0 of 10 | 0 of 10 | 0 of 10 |
| TPC Freq. | 1 in 159 | NA | NA | 1 in 4840 | TPC Freq. | 1 in 530 | 1 in 1080 | NA | NA |
| 95% CI | 63–401 | NA | NA | 632–37094 | 95% CI | 194–1445 | 395–2957 | NA | NA |
| Cumulative TPC frequency kRASG12D | | | | | Cumulative TPC frequency kRAS^{G12D} + mylpfa:myf5 | | | | |
| Cell # | G^+ | G^+R^+ | R^+ | DN | Cell # | G^+ | G^+R^+ | R^+ | DN |
| TPC Freq. | 1 in 146 | 1 in 4206 | NA | NA | TPC Freq. | 1 in 377 | 1 in 639* | NA | NA |
| 95% CI | 87–245 | 1550–11409 | NA | NA | 95% CI | 212–670 | 363–1125 | NA | NA |

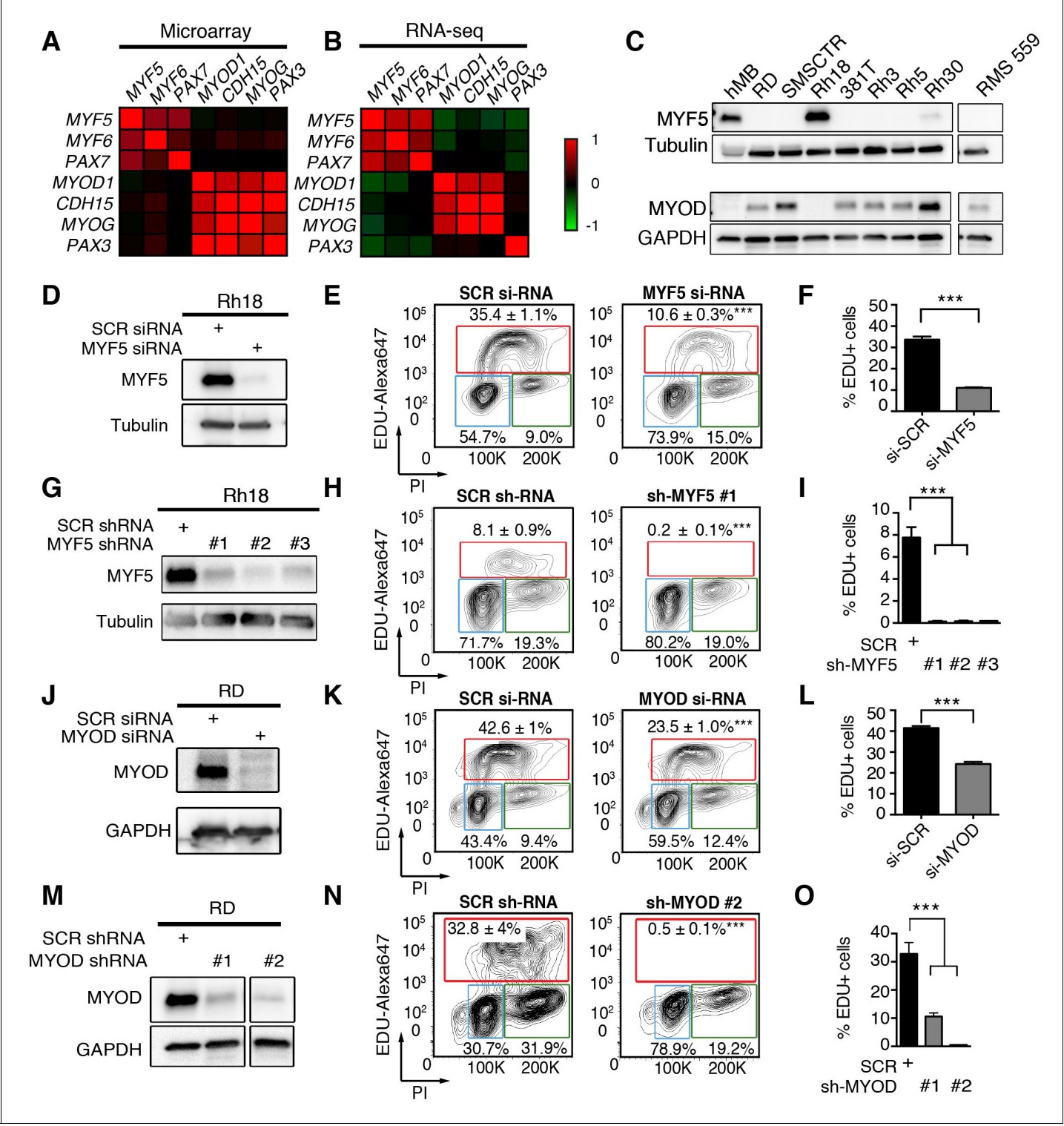

**Figure 4.** MYF5 and MYOD are required for human ERMS proliferation and growth. (A–B) Pearson correlation for gene expression of myogenic genes in primary human RMS as assessed by microarray (A) or RNA-sequencing (B). Heatmap represents correlation coefficients. (C) Western blot analysis for MYF5 and MYOD in human RMS cell lines. (D–I) Rh18 ERMS cells following MYF5 knockdown with siRNA (D–F) or shRNA (G–I). (J–O) RD ERMS cells following MYOD knockdown with siRNA (J–L) or shRNA (M–O). Western blot analysis following knockdown at 48 hr (D,J) and 72 hr (G,M). EdU and Propidium Iodide (PI) cell cycle analysis assessed by flow cytometry at 48 hr (E,F,K,L) and 72 hr (H,I,N,O). Standard Deviation denoted in FACS plots and graphs. Analysis shown in D-O was completed as technical replicates and completed ≥3 independent times with similar results. ***p<0.001 by Student's t-test.

*Figure 4 continued on next page*

*Figure 4 continued*

The following figure supplements are available for figure 4:

**Figure supplement 1.** Pearson correlation of gene expression from RNA-seq data of primary human RMS.

**Figure supplement 2.** Immunofluorescence for MYF5 and MYOD in Rh18 and RD ERMS cell lines.

**Figure supplement 3.** MYF5 and MYOD are required for human RMS proliferation and growth in vitro.

**Figure supplement 4.** MYF5 and MYOD are each specifically required for human RMS proliferation and growth in vitro.

was reduced >30% following stable sh-*MYOD* knockdown (*Figure 4—figure supplement 3H*). In support of MYOD having important roles in regulating TPC number, sphere colony formation was also greatly reduced in RD cells following shRNA knockdown (p<0.01, *Figure 4—figure supplement 3I,J*). Sphere colony formation is an in vitro surrogate for quantifying TPC number and correlates well with in vivo limiting dilution cell transplantation experiments (*Walter et al., 2011*; *Satheesha et al., 2016*). Finally, *MYOD* knockdown also impaired cell cycle and growth in additional RMS cell lines including ERMS cell lines 381T and RMS559 and the ARMS cell line Rh3. These cells lines all had significant reductions in S-phase cycling cells following siRNA treatment with variable effects on apoptosis at the time points analyzed (*Figure 4—figure supplement 4A–L*). Importantly, specificity of MYF5 and MYOD knockdown was confirmed in multiple cells lines, showing that siRNA knockdown effects were specific to each MRF and that myogenic factors were not redundantly re-expressed following knockdown (*Figure 4—figure supplement 4M–T*). In total, our data show that MYF5 and MYOD are individually expressed in different RMS tumor cells and yet have similar roles in regulating cell cycle progression and proliferation in RMS.

## Myogenic transcription factors are required for continued xenograft growth

Given the prominent role *MYF5* had in regulating cell growth in human ERMS cells in vitro and imparting tumor propagating potential to differentiated zebrafish ERMS cells, we next wanted to assess if MYF5 was required for ERMS maintenance and growth in vivo. Rh18 cells were infected with shRNAs and harvested at 72 hr post-infection. *MYF5* knockdown was confirmed by Western blot analysis (*Figure 5A*). *Luciferase-mKate* expressing Rh18 shRNA cells were transplanted into the flanks of *NOD/SCID/IL2rg* null mice ($1 \times 10^6$ viable cells in matrigel per site). Non-targeting control shRNA cells were implanted subcutaneously into the left flank and *MYF5* knockdown cells into the right (N = 6 animals, two independent shRNAs). 5 hr after injection, mice were injected with luciferin and bioluminescence was measured, confirming that the same amount of control and knockdown cells had been injected into recipient mice (*Figure 5B*). Serial bioluminescence imaging showed that tumor volume was reduced in *MYF5* knockdown cells while control cells continued to grow (p<0.05, Student's t-test; *Figure 5—figure supplement 1A*). *MYF5* knockdown cells were largely undetected at late time points (p<0.05, Student's t-Test; *Figure 5B,C*). Analysis of mice at necropsy revealed that only 2 of 12 mice had tumors derived from MYF5-deficient Rh18 cells and that overall tumor weight was greatly reduced when compared with control Rh18 knockdown cells (p<0.01; Student's t-test; *Figure 5D–E*). For these rare ERMS that developed from MYF5 knockdown Rh18 cells, they retained ERMS histology (*Figure 5—figure supplement 1D,E*). Taken together, these data show that MYF5 is required for efficient xenograft tumor cell growth in vivo.

Next, we assessed if MYOD was also important for continued xenograft growth of human ERMS cells. Using the same approach outlined above, we found that shRNA knockdown of MYOD resulted in efficient knockdown prior to tumor cell implantation (*Figure 5F,K*) and reduced xenograft growth of both RD and RMS559 cells when assessed by total body luciferase imaging completed overtime (p<0.01; *Figure 5F–H,K–M* and *Figure 5—figure supplement 1B,C*). MYOD knockdown tumors were also smaller at the time of necropsy (*Figure 5I,N*) and weighed significantly less than control shRNA expressing tumors (p<0.01, Student's t-test; *Figure 5J,O*). Unlike our Rh18 experiments, *MYOD* knockdown cells continued to grow in transplant recipient animals, albeit at greatly reduced

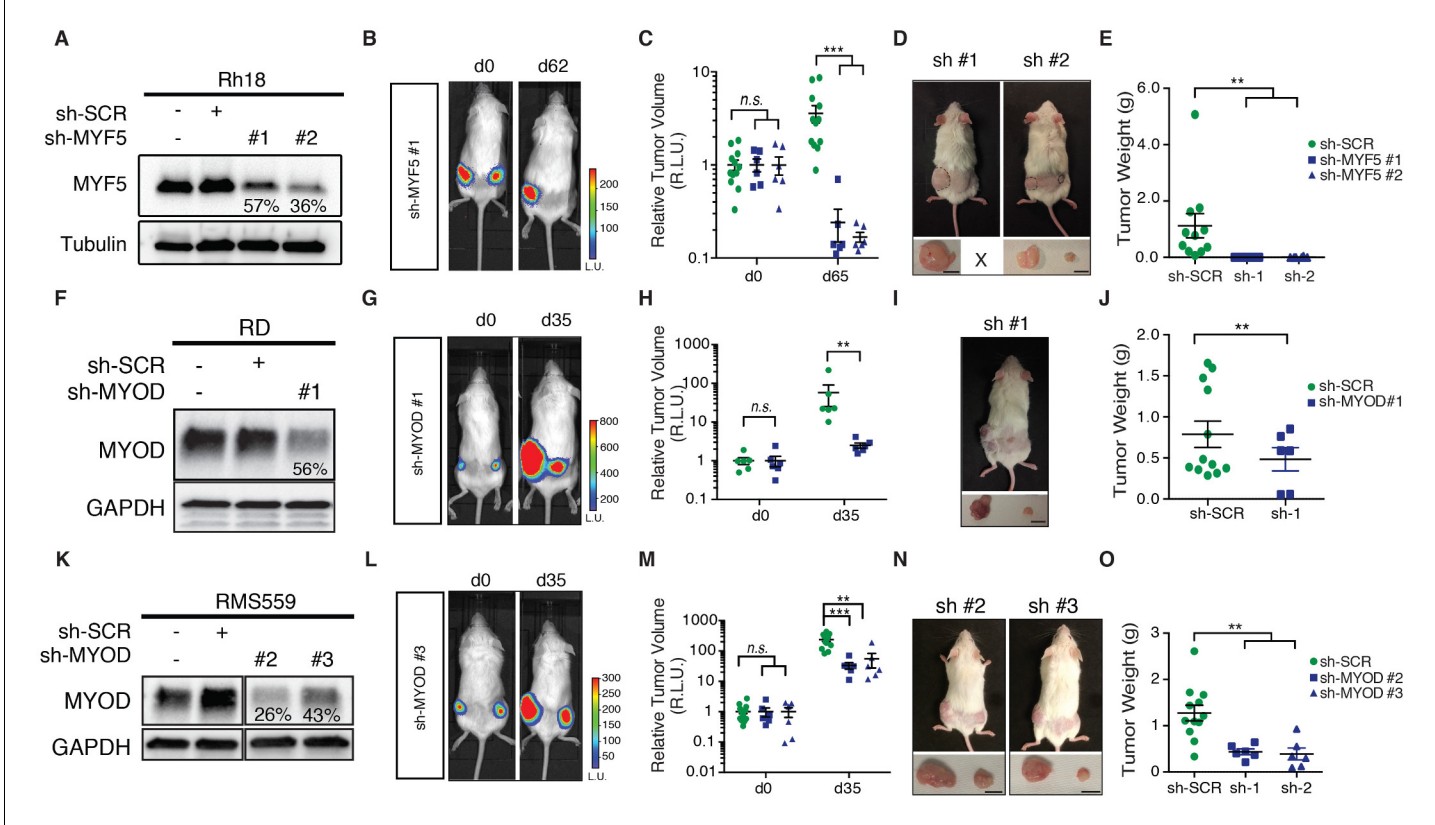

**Figure 5.** MYF5 and MYOD are required for human ERMS xenograft growth. Xenograft growth in Rh18 (**A–E**), RD (**F–J**), and RMS559 (**K–O**) following knockdown with scramble control shRNA (sh-SCR) or shRNAs specific to MYF5 or MYOD. (**A,F,K**) Western blot analysis of shRNA expressing cells harvested for transplantation at 72 hr after lenti-viral shRNA knockdown. Percent knockdown compared to shRNA control is shown. (**B,G,L**) Luciferase bioluminescent imaging of a representative animal at the time of implantation (left panel) or at later time points (right panel). Control shRNA cells were implanted into left flank and knockdown cells into the right (N = 6 mice per shRNA). Intensity represents total luminescence units measured per region of interest (L.U.) (**C,H,M**) Quantification tumor volume when assessed by luciferase imaging. Relative luminescence units (R.L.U.). (**D,I,N**) Representative images of mice at the time of necropsy, with excised tumors shown in lower panels. (**E,J,O**) Quantification of tumor weight at the time of necropsy. Tumors that could not be identified at time of necropsy were assigned a value of zero for this analysis. Standard Error of the Mean are denoted in graphs. **p<0.01; ***p<0.001 by Mann-Whitney non-parametric test. Scale bar equals 1 cm in D,I, and N.

The following figure supplement is available for figure 5:

**Figure supplement 1.** MYF5 and MYOD are required for human ERMS growth and maintenance following xenograft transplantation into *NOD/SCID/IL2g* null mice.

---

levels. These data suggest both important similarities for MRF factors in driving proliferation and growth and yet, also suggest that additional molecular mechanisms likely contribute to continued tumor growth in MYOD-expressing ERMS. As with the Rh18 experiments, tumors that formed following *shMYOD* knockdown retained similar RMS morphology when compared with control treated cells (*Figure 5—figure supplement 1F–I*).

### *MYF5* and *MYOD* control common transcriptional targets to regulate proliferation and myogenic state in human RMS

Our work uncovered that MYF5 and MYOD are largely mutually exclusively expressed in RMS and that each is required for proliferation. MYF5 and MYOD also bind directly to enhancers of well-known muscle regulated genes in development, including myogenin and *CDH15* (m-cadherin) (*Conerly et al., 2016*). These data suggest that these transcription factors likely regulate a common set of transcriptional targets that lock RMS cells in a proliferative myogenic state. To further explore this hypothesis, we next performed ChIP-seq for MYF5 in Rh18 cells and compared these results

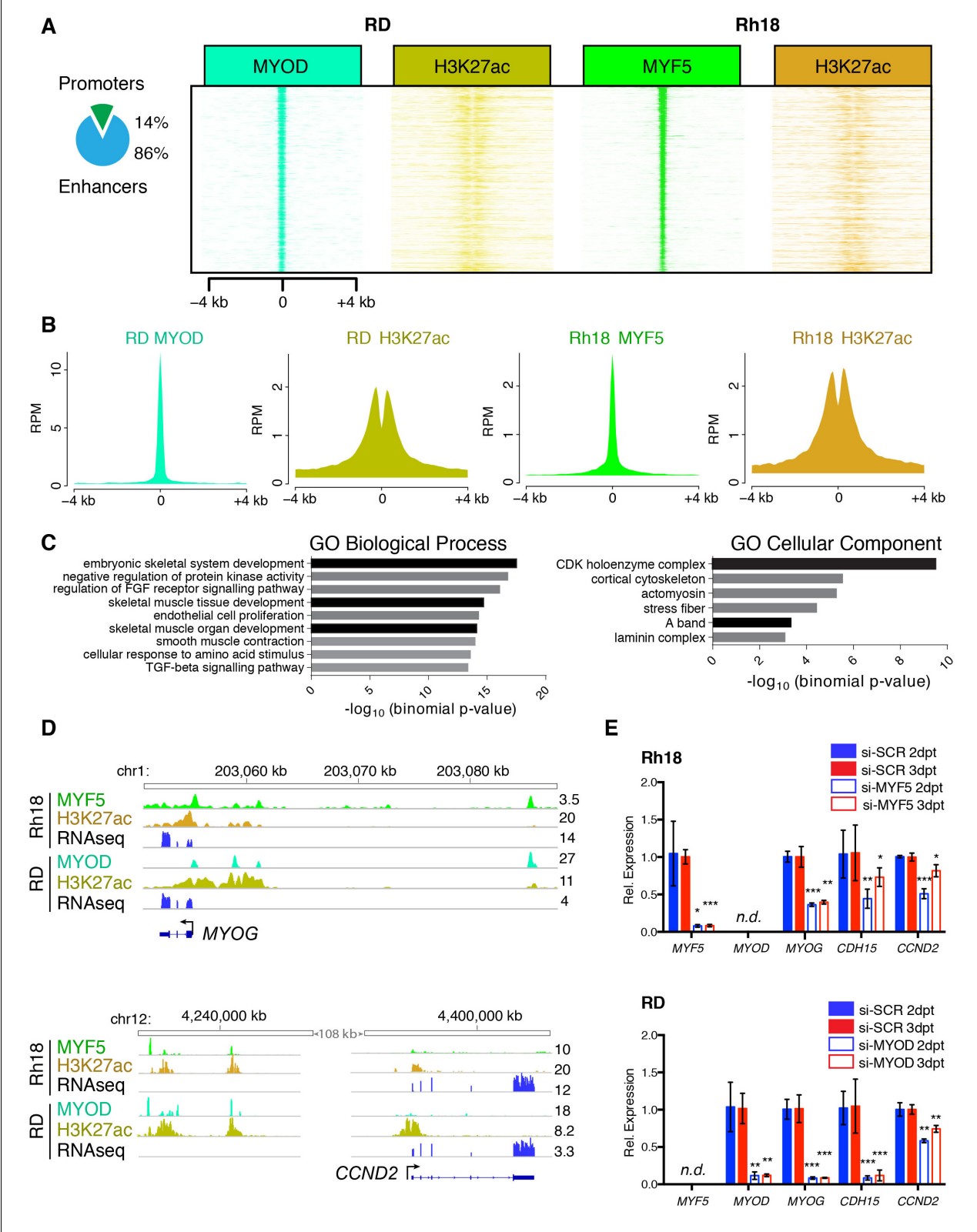

**Figure 6.** MYF5 and MYOD bind common promoter and enhancer regions and induce genes involved in muscle development and cell cycle. (A–B) ChIP-seq analysis showing genomic regions bound by both MYOD in RD cells and MYF5 in RH18 cells. H3K27 acetylation (H3K27ac). (C) Gene ontology enrichment of gene regions bound by both MYOD in RD cells and MYF5 in RH18 cells. GO Biological Processes, GO Cellular Component predictions, and binomial p-values denoted. (D) Signal tracks for ChIP-seq and RNA-seq surrounding *MYOG* (top) and *CCND2* (bottom). Numbers to the right

*Figure 6 continued on next page*

Tenente *et al*. eLife 2017;6:e19214. DOI: 10.7554/eLife.19214

*Figure 6 continued*

indicate reads per million mapped reads. (**E**) Quantitative real-time PCR gene expression analysis of RH18 (top) and RD cells (bottom). Cells were assessed following siRNA-mediated knockdown at 2 days (2dpt, blue bars) or 3 days post-transfection (three dpt, red bars). Error bars denote standard deviation. Student's t-test; *p<0.05, **p<0.01, ***p<0.001.

The following figure supplements are available for figure 6:

**Figure supplement 1.** MYF5 and MYOD bind common promoter and enhancer regions.

**Figure supplement 2.** *Ccnd2a* expression in zebrafish ERMS.

with ChIP-seq data performed for MYOD in RD cells (*MacQuarrie et al., 2013a*) (*Figure 6* and *Figure 6—figure supplement 1*). This analysis uncovered a common set of promoter and enhancer regions bound by both MYOD and MYF5 (*Figure 6A*). 86% of commonly bound genomic DNA regions were confined to enhancer regions as defined by H3K27-acetylation occupancy (*Figure 6A, B*). Unbiased analysis of commonly bound target genes using GREAT (*McLean et al., 2010*), *Supplementary file 2*) revealed an enrichment of genes that regulate cell cycle and myogenic cell fate (*Figure 6C*). Enrichment of GO terms included 'cyclin-dependent protein kinase holoenzyme complex' (*Supplementary file 3*), 'skeletal muscle tissue development' and 'embryonic skeletal system development' (*Supplementary file 4*; binomial, $p<1\times10^{-9}$). Signal tracks of ChIP-seq and RNA-seq independently confirmed common binding of MYF5 and MYOD to genes that regulate cell cycle and myogenic cell fate (*Figure 6D*), including *cyclin-dependent kinase cyclin D2* (*CCND2*), m*yogenin* (*MYOG*), and *cadherin 15 (m-Cadherin, CDH15*, *Figure 6D* and *Figure 6—figure supplement 1B*).

In order to show that MYF5 and MYOD are regulators of cell fate, we next performed qRT-PCR gene expression analysis for muscle differentiation genes following *si-MYF5* or *si-MYOD* knockdown in Rh18 and RD cells, respectively (*Figure 6E*). As expected if myogenic transcription factors regulate muscle cell fate, both *Myogenin* and *CDH15* were downregulated following MYF5 or MYOD knockdown at either 48 or 72 hr (*Figure 6E*). Additionally, the cell cycle regulatory gene, *CCND2* was also reduced following MYF5 or MYOD knockdown (*Figure 6E*). Moreover, the zebrafish orthologue *ccnd2a* transcript showed a trend toward higher expression in *mylpfa:myf5* expressing zebrafish ERMS (*Figure 6—figure supplement 2*), correlating well with elevated *myf5* expression in these tumors. *CCND2* is a CDK4/6-associated cyclin that is amplified in a subset of human RMS and is required for cell proliferation and viability in human RMS (*Chen et al., 2013a*). Importantly, CCND2 is a predicted direct target of MYF5 and MYOD binding (*Conerly et al., 2016*). Together, our data show that MYF5 and MYOD regulate common gene programs that lock cells in an arrested myogenic fate and are required for sustained proliferation of RMS cells.

## Discussion

Rhabdomyosarcomas express b-Helix-loop-Helix (bHLH) myogenic regulatory transcription factors (MRFs), including MYF5 and MYOD (*Clark et al., 1991*; *Kumar et al., 2000*; *Sebire and Malone, 2003*) but fail to activate terminal muscle differentiation programs. Several mechanisms have been shown to play a role in this differentiation arrest. These include disruption of the balance of MRF-E12 heterodimers and inhibitor complexes (*Macquarrie et al., 2013b*; *Yang et al., 2009*), presence of inhibitory miRNAs (*Macquarrie et al., 2012*), and deregulation of cell cycle (*Fiddler et al., 1996*). These data have therefore led to the suggestion that MRFs do not have a role in RMS transformation or in sustained tumor growth, but are merely retained from the target cell of transformation (*Keller and Guttridge, 2013*). While *MYOD* overexpression fails to differentiate ERMS cells (*Yang et al., 2009*), both MYF5 and MYOD potently reprogram fibroblasts into proliferating muscle cells (*Braun et al., 1989*; *Tapscott et al., 1988*). Moreover, MYF5 and MYOD are commonly re-expressed in experimental animal models of RMS irrespective of the cell of origin (*Hettmer et al., 2011, 2015*; *Ignatius et al., 2012*; *Langenau et al., 2007*; *Rubin et al., 2011*; *Storer et al., 2013*) suggesting roles for these transcription factors in driving tumor growth and TPC function. Our experiments have shown that Myf5 can impart tumor-propagating potential to ERMS cells in the zebrafish model, suggesting important roles for myogenic regulatory transcription factors in

regulating growth. These data were confirmed by loss-of-function studies in human RMS where MYF5 or MYOD knockdown suppressed RMS proliferation and reduced viability in vitro. Similar effects were also observed in xenograft studies, where MYF5- and MYOD-knockdown ERMS cells failed to grow efficiently in vivo. Remarkably, despite over 25 years of study into MYOD and MYF5, loss-of-function studies in RMS have not been reported and thus roles for these factors in regulating RMS growth have gone unappreciated.

It is becoming increasingly noted that developmental transcription factors and pathways are commonly co-opted by cancer to regulate growth and tumor-propagating activity. For example, the TAL1/SCL bHLH transcription factor is required for hematopoietic stem cell specification and self-renewal during development. TAL1/SCL is overexpressed in 60% of T-cell acute lymphoblastic leukemia (T-ALL) (*Ferrando et al., 2002*) and can reprogram thymocytes into self-renewing, pre-leukemic cells (*Gerby et al., 2014*). This same paradigm has also been seen in brain tumors. For example, the bHLH transcription factor OLIG2 is required for self-renewal of normal neural progenitor cells (*Imayoshi and Kageyama, 2014*) and is also a marker of glioblastoma TPCs (*Beyeler et al., 2014*; *Ligon et al., 2007*; *Suvà et al., 2014*; *Trépant et al., 2015*). Our results in RMS parallel those outlined for T-ALL and glioblastoma, showing that MRFs are capable of reprogramming differentiated RMS cells into TPCs and are required for sustained proliferation and viability of human RMS. Our data also suggests, that like T-ALL and glioblastoma, the MYF5 and MYOD bHLH proteins regulate common molecular pathways in self-renewal and growth of both normal and malignant muscle. Importantly, transcription factors have recently been therapeutically targeted (*Bhagwat and Vakoc, 2015*; *Roe et al., 2015*), raising hope that developing drugs that inhibit MYF5 and MYOD cancer cell dependencies could be efficacious in treating RMS patients in the future.

Our molecular analysis also uncovered that MYF5 and MYOD are mutually-exclusively expressed in human RMS. Loss-of-function studies uncovered important roles for either MYF5 or MYOD in regulating RMS growth and muscle cell fate. This data contrasts starkly with co-expression of these factors and functionally overlapping roles in development. For example, MYF5 and MYOD are well-known to act redundantly in muscle development to regulate muscle specification and differentiation (*Rudnicki et al., 1993*). This same redundancy of Myf5 and MyoD in development and muscle injury has now been reported in zebrafish (*Hinits et al., 2009*; *Siegel et al., 2013*), *Drosophila* (*Abmayr and Keller, 1998*) and *Xenopus* (*Chanoine and Hardy, 2003*), showing a high conservation of the MYOD/MYF5 transcriptional machinery in regulating muscle specification and development throughout evolution. These same MRFs are also required for self-renewal of adult muscle satellite cells (*Cooper et al., 1999*; *Ustanina et al., 2007*; *Yablonka-Reuveni et al., 1999*), yet roles for these factors in individually regulating muscle fate and self-renewal are now just emerging in the literature. For example, a subset of muscle progenitors are specified by MYOD without the contribution of MYF5 (*Haldar et al., 2008*, *2014*). Moreover, a subset of adult muscle progenitors express MYF5 and then MYOD sequentially during their specification with both being required for muscle regeneration following injury (*Comai et al., 2014*). Our data suggest that either MYOD or MYF5 are uniquely expressed within human RMS and are each individually sufficient to drive tumor growth. These data are consistent with similar roles for either Myf5 or MyoD to reprogram fibroblasts into muscle cell fates (*Braun et al., 1989*; *Tapscott et al., 1988*) and a high degree of overlap in binding of common enhancer and promoter targets with normal myoblasts (*Conerly et al., 2016*).

Finally, our work uncovered downstream pathways required for RMS growth that were regulated by MYOD and MYF5, including muscle specification programs and cell cycle. For example, *m-cadherin* (*CDH15*) and *myogenin* (*MYOG*) were transcriptionally regulated by MYF5 and MYOD in human RMS, consistent with regulation of these same factors by MYF5 and MYOD in myoblasts (*Conerly et al., 2016*). Our work also uncovered roles for MRFs in regulating *cyclin D2* (*CCND2*) in human RMS. Importantly, the CDK4/6-associated cyclin D2 (*CCND2*) complex is required for both myoblast proliferation and human RMS growth, is highly expressed in primary human RMS and is amplified in a subset of human and zebrafish RMS (*Chen et al., 2013a*; *Saab et al., 2006*; *Webster and Fan, 2013*). Collectively, our data suggest that MYOD and MYF5 likely exert import roles in regulating muscle cell identity and cell cycle regulation, both of which are required for sustained tumor growth and likely shared with normal muscle to regulate stem cell self-renewal.

## Materials and methods

### Animals and protocol approvals

Studies were approved by the Massachusetts General Hospital Subcommittee on Research Animal Care under the protocol #2011 N000127 (zebrafish) and #2013 N000038 (mouse). Biosafety lentiviral work was approved by the Partners IBC under protocol #2013B000039. Zebrafish used in this work include: CG1 strain (*Mizgireuv and Revskoy, 2006*), *myf5*-GFP (*Chen et al., 2007*) and *mylpfa*-mCherry (previously *mylz2*-mCherry)(*Xu et al., 1999*) transgenic zebrafish lines and *rag*$^{E450fs}$ (ZFIN IND *rag2*$^{fb101}$) homozygous fish (*Tang et al., 2014*; *Tenente et al., 2014*). *myf5*-GFP/*mylpfa*-mCherry double transgenic fish (AB strain) were outcrossed 10 times into CG1-strain zebrafish to generate compound syngeneic transgenic zebrafish (*Ignatius et al., 2012*). 6-week-old *NOD/SCID/Il2rg* null female mice were used in this work.

### Micro-injection and ERMS generation in transgenic zebrafish

*rag2-kRAS*$^{G12D}$ and *mylpfa-mCherry* constructs were described previously (*Langenau et al., 2007*; *Smith et al., 2010*). The *mylpfa-myf5* construct was obtained by gateway cloning using a zebrafish *myf5* ORF from 24hpf zebrafish embryo cDNA (http://tol2kit.genetics.utah.edu). *rag2-kRAS*$^{G12D}$ and *mylpfa-myf5* constructs were linearized with XhoI, phenol:chloroform-extracted, ethanol-precipitated, re-suspended in 0.5× Tris-EDTA + 0.1 M KCl, and injected into one-cell stage embryos of the respective backgrounds, as previously described (*Langenau et al., 2007*; *Tenente et al., 2014*). We and others have used the *mylpfa* promoter to drive transgene expression in differentiated muscle cells both in the stable and mosaic transgenic setting (*Ju et al., 2003*; *Ignatius et al., 2012*; *Storer et al., 2013*; *Tang et al., 2016*) confirming that mylpfa transgene expression is confined to differentiated ERMS cells.

### Quantification of zebrafish RMS size, tumor onset, and penetrance

Zebrafish were monitored every 3–4 days for time-to-tumor onset using an epi-fluorescent stereomicroscope. Animals were imaged at 10 days postfertilization until 55 days postfertilization. Primary tumor size was quantified from 6.3x or 10x photomicrographs taken at 30 postfertilization and calculated by multiplying fluorescence intensity by 2D pixel area using the ImageJ software package as previously described (*Chen et al., 2014*). Kaplan-Meier tumor onset and penetrance analysis was performed using Graphpad Prism Software and statistically analyzed using the Log-rank statistic.

### Zebrafish histology, immunohistochemistry and EdU incorporation

Paraffin embedding, sectioning and immunohistochemical analysis of zebrafish sections were performed as described (*Chen et al., 2013a*, *2014*; *Ignatius et al., 2012*). Antibodies used for immunohistochemistry included: phospho-H3 (1:6000, Santa Cruz Biotechnology, Dallas, Texas) and cleaved-caspase3 (CC3, 1:250, Cell Signaling Technology, Danvers, MA). All histopathology procedures were performed at the MGH and BWH DF/HCC Research Pathology Cores. Slides were imaged using a transmitted light Olympus BX41 microscope. Pathology review and staging were completed by G.P. N. Tumor histology classification was assigned as described in *Figure 1—figure supplement 2* and *Figure 2—figure supplement 1* with stage one being the least differentiated with tumors being comprised of only small round blue cells. Stage 2 and 3 ERMS were assigned based on the preponderance of rhabdomyoblast cells, fibrous and spindle cell morphology, with a low proportion of interspersed smaller round blue cells. EdU was injected intraperitoneally into live tumor-bearing zebrafish and incubated for 6 hr prior to fixation as described previously (*Ignatius et al., 2012*). Animals were cryosectioned and stained using the Click-iT Alexa Fluor 647 imaging kit (Invitrogen, Carlsbad, CA). Images were acquired using a Zeiss 710 Confocal microscope (Zeiss, Oberkochen, Germany).

### Zebrafish ERMS cell transplantation and FACS

FACS analysis and RMS cell transplantation by intra-peritoneal injection were completed essentially as described (*Chen et al., 2014*; *Ignatius et al., 2012*; *Langenau et al., 2007*; *Smith et al., 2010*). Freshly isolated RMS tumor cells were stained with DAPI to exclude dead cells and sorted twice using a Laser BD FACSAria II Cell Sorter. Sort purity and viability were assessed after two rounds of

sorting when possible, exceeding 85% and 95% respectively. Fish were monitored for tumor engraft-ment from 10 to 120 days post transplantation. Tumor-propagating cell frequency was quantified following transplantation into CG1 syngeneic recipient fish using the Extreme Limiting Dilution Anal-ysis software (http://bioinf.wehi.edu.au/software/elda/). A subset of transplanted fish were fixed in 4% PFA in PBS, sectioned, stained with Hematoxylin and Eosin (H and E), and staged for differentia-tion score.

## Gene expression analysis

Total RNA was isolated from AB-strain embryos 6 and 24 hr postfertilization, FAC-sorted ERMS cell subpopulations, bulk unsorted primary zebrafish ERMS or human RMS samples. Quantitative real-time PCR utilized gene-specific PCR primers (*Supplementary file 1*), and expression was normalized to 18S controls (zebrafish samples) or *GAPDH* (human samples) to obtain relative transcript levels using the $2^{-ddCT}$ method. Technical triplicates were completed for all qRT-PCR reactions and data presented as average expression ±1 standard deviation. For zebrafish RMS sub-populations, relative gene expression was normalized within individual samples, and cumulative transcript expression across the two ERMS cell subpopulations was set to 50. Samples were assessed in relation to 24 hr postfertilization embryos to ensure that results for $2^{-ddCT}$ results for any given gene were not lower than 10-fold expression found in normal development, as previously described (*Ignatius et al., 2012*). For zebrafish bulk primary ERMS gene expression analysis, samples were also assessed in relation to 24 hr postfertilization embryos. For human RMS gene expression analysis, knockdown samples were assessed in relation to si-SCR controls.

## Human RMS cell lines

The human RD cell line was obtained from ATCC's cell biology collection (Manassas, Virginia). SMS-CTR, 381T, Rh3, Rh5 and Rh30 cell lines were kindly provided by Dr. Corrine Linardic (Duke Univer-sity, North Carolina), the Rh18 cell line (fusion-negative) by Dr. Peter Houghton (Ohio State Univer-sity, now at UTHSCSA) and RMS559 by Dr. Jonathan Fletcher (Brigham and Women's Hospital, Massachusetts). All RMS cell lines were authenticated by STR profiling and were mycoplasma tested. Cell lines used in this work are not commonly misidentified based on the International Cell Line Authentication Committee. The human MB1208-1 human skeletal myoblast cell line was kindly pro-vided by Dr. Louis Kunkel (Boston Children's Hospital, Massachusetts). Characteristics of these human RMS (*Hinson et al., 2013*; *Sokolowski et al., 2014*) and skeletal myoblast (*Alexander et al., 2011*) cell lines have been reported previously.

## Western blot analysis

Total cell lysates from human RMS cell lines were obtained following lysis in 2%SDS lysis buffer sup-plemented with protease inhibitors (Santa Cruz Biotechnology, Dallas, Texas). Samples were boiled, vortexed and homogenized through a 28G syringe. 20–40 µg of protein was loaded in 4–20% Mini-Protean TGX gels (Biorad, Hercules, CA) and transferred onto PVDF membranes. Western blot analy-sis used primary antibodies: rabbit a-MYF5 (1:5000, Abcam ab125078, Cambridge, MA), mouse a-MYOD1 (1:1000, Abcam ab16148, Danvers, MA), rabbit a-MYOD1 (1:1000, Abcam ab133627), rabbit a-GAPDH (1:2000, Cell Signaling 2118), mouse a-TUBULIN (1:2500, Abcam ab4074) and sec-ondary antibodies: HRP anti-rabbit (1:2000, Cell Signaling 7074) or HRP anti-mouse (1:3000, GE Healthcare NA93IV, Marlborough, MA). Blocking was completed using 5% skim milk/TBST. Mem-branes were developed using an ECL reagent (Western Lightening Plus-ECL, Perkin Elmer, Waltham, MA or sensitive SuperSignal West phemto Maximum Sensitivity Substrate, Thermo Scientific, Waltham, MA).

## *MYF5* and *MYOD* siRNA knockdown and immunofluorescence

Gene-specific smart-pool or control siRNAs (Dharmacon, GE Life Sciences, Marlborough, MA) (0.01 µM) were reverse-transfected into cells using RNAiMax lipofectamine transfection reagent (Life Technologies, Waltham, MA) in flat clear bottom 96 well plates. Cells were then fixed at 72 hr post transfection in 4% PFA/PBS, washed in x1 PBS and permeabilized in 0.5% TritonX-100/PBS. Antibod-ies used were rabbit a-Myf5 (1:400, Abcam ab125078) and mouse a-MyoD (1:200, Abcam ab16148) in 2% goat serum/PBS, Alexa 488 goat anti-mouse (1:1000, Invitrogen A11029) and Alexa 594 goat

anti-rabbit (1:1000 Invitrogen A11037). Cells were incubated with DAPI (1 µg/ml), and imaged at 200x using a LSM710 Zeiss Laser scanning confocal microscope. Images were processed in ImageJ and Adobe Photoshop.

For EdU and AnnexinV assays, gene-specific smart-pool or control siRNAs (Dharmacon, GE Life Sciences) (5 µM) were added to Rh18, RD, 381T, RMS559 and Rh3 cells in a 6-well plate and incubated for 48–96 hr prior to analysis.

## *MYF5* and *MYOD* lentiviral shRNA knockdown

Non-targeting scrambled (SCR) control shRNA and *MYF5* or *MYOD* specific shRNAs were delivered on the pLKO.1-background vector (from MGH Molecular Profiling Laboratory) and packaged using 293T cells (*Supplementary file 1*). RMS cells were infected with viral particles for 24 hr at 37°C with 8 µg/ml of polybrene (EMD Millipore, Billerica, MA).

## Human RMS in vitro assays

Nuclei counts were performed following incubation with NucBlue Live ReadyProbes Reagent (Life Technologies). Cells were imaged at 100X and 400X magnification using an inverted fluorescent microscope. Manual cell counts were performed using the ImageJ software. Three fields were counted per well and completed in triplicate. Cell cycle analysis was performed using the EdU Click-iT plus EdU Flow Cytometry-AlexaFluor 647 picol azide assay (Life Technologies) following 2 hr (RD, 381T, Rh3, RMS559) or 6 hr incubation (Rh18) with 10 µM EdU. Apoptosis was assessed using the AnnexinV-AlexaFluor 647/PI or 7AAD assays (Life Technologies and BD Biosciences, San Jose, CA). Flow cytometric analysis was performed using the SORP4 Laser BD LSRII Flow Cytometer and processed with the FlowJo Software. All experiments were run as technical triplicates and repeated ≥3 independent times. Sphere formation assays using RD cells were completed essentially as previously described (*Walter et al., 2011*).

## Primary human RMS gene expression and correlation analysis

Previously published microarray gene expression data were processed and normalized using Robust Multichip Average (RMA) normalization (*Davicioni et al., 2006*) (raw data was obtained from the NCI Cancer Array Database). Previously published RNA-seq gene expression data from human RMS were processed and normalized using a standard Tuxedo pipeline (*Shern et al., 2014*; *Trapnell et al., 2012*). The resulting expression values from the microarray and RNA-seq datasets were then log2 transformed. Pearson correlation was determined for the following genes: *CDH15*, *MYF5*, *MYF6*, *MYOD1*, *MYOG*, *PAX3*, and *PAX7*. The correlation heatmap was plotted using the R package 'fheatmap' (Fantastic Heatmap. R package version 1.0.1. http://CRAN.R-project.org/package=fheatmap) and processed using Adobe Photoshop.

## ChIP-seq of human RMS cell lines

Chromatin immunoprecipitations were performed on Rh18 cells using the Chip-IT High sensitivity kit (Active Motif, Carlsbad, CA) and anti-H3K27ac (Active Motif) or anti-MYF5 (C-20, Santa Cruz) antibodies. Resultant purified immune-precipitated DNA was used for library preparation using the Tru-Seq ChIP sample preparation kit (Illumina, San Diego ,CA) without modifications. 11–18 library preps were mixed for multiplexed single read sequencing using the NextSeq500 (Illumina).Previously published MYOD ChIP-seq data from RD was downloaded and processed in parallel with the newly generated sequencing data (GSE50415, GSE84630) (*MacQuarrie et al., 2013a*). Reads were aligned to the hg19 reference using BWA. ChIP-seq peaks were identified using MACS 2.1 (*Zhang et al., 2008*). Gene ontology was performed using GREAT (*McLean et al., 2010*). Differential peak calling between RD and Rh18 was performed using bedtools v2.25.0 and visualized using NGS plot (*Shen et al., 2014*). Genomic regions were visualized using IGV v2.3.40.

## Mouse xenografts, luciferase imaging, necropsy and histological analysis

Rh18, RD and RMS559 ERMS cells were co-infected with pLKO.1-shRNA lentivirus and pLKO.1-*luc-mKate* (gift from Drs. Matthijssens and Van Vlierberghe, Ghent University, Belgium). At 3 days post-infection, cells were collected and counted. An aliquot of cells was analyzed using the SORP4 Laser

BD LSRII Flow Cytometer to determine viability following DAPI staining. Separate aliquots of cells were harvested and used for Western blot analysis. Equal numbers of viable cells were then embedded into Matrigel (Corning Life Sciences, Tewksburg ,MA) at a final concentration of $1 \times 10^6$ of viable cells per 200 µl. Six-week-old *NOD/SCID/IL2rg* null female mice were anesthetized by isofluorane and transplanted with scramble-shRNA/mKate-luc cells subcutaneously into the left flank (N = 6 animals/shRNA construct) whereas sh-MYF5 or sh-MYOD/mKate-luc cells were injected on the right (200 µl/flank injection). Tumor growth was monitored by bioluminescence imaging following subcutaneous injection into the loose tissue over the neck of 75 mg/kg D-luciferin (Perkin Elmer, Waltham, MA) in 100 µl of PBS. Imaging was completed and analyzed using the IVIS Lumina II (Caliper Life Science, Hopkinton, MA). At time of necropsy, mouse brightfield images were acquired using a regular camera and tumors were excised, weighed, processed and stained using Hematoxylin and Eosin. Comparisons of tumor size and weight between groups used the Student's t-test after a normality test was performed, otherwise a Mann-Whitney analysis was performed, as indicated in the figure legends.

## Acknowledgements

This work was funded by NIH grants R01CA154923 (DML), R24OD016761 (DML), U54CA168512 (DML), and a St. Baldricks Research Grant (DML). We thank the Specialized Histopathology Services at Massachusetts General Hospital (MGH) and the Dana-Farber/Harvard Cancer Center (P30 CA06516), the MGH Cancer Center/Molecular Pathology Confocal Core and the MGH CNY Flow Cytometry Core and Flow Image Analysis (1S10RR023440-01A1). We thank Matthew Alexander, PhD for helpful suggestions and comments.

## Additional information

### Funding

| Funder | Grant reference number | Author |
|---|---|---|
| National Institutes of Health | | Inês M Tenente |
| Fundação para a Ciência e a Tecnologia | | Myron S Ignatius<br>David M Langenau |
| China Scholarship Council | | Qin Tang |
| Alex's Lemonade Stand Foundation for Childhood Cancer | | David M Langenau |
| Saint Baldrick's Foundation | | David M Langenau |
| National Institutes of Health | R01CA154923 | David M Langenau |
| National Institutes of Health | U54CA168512 | David M Langenau |
| NIH Office of the Director | R24OD016761 | David M Langenau |

The funders had no role in study design, data collection and interpretation, or the decision to submit the work for publication.

### Author contributions

IMT, MNH, MSI, KM, MLO, JK, Conception and design, Acquisition of data, Analysis and interpretation of data, Drafting or revising the article; MY, EYC, Conception and design, Acquisition of data, Drafting or revising the article; SS, Analysis and interpretation of data, Drafting or revising the article; BG, DML, Conception and design, Analysis and interpretation of data, Drafting or revising the article; AR, QT, GPN, Acquisition of data, Analysis and interpretation of data, Drafting or revising the article

### Author ORCIDs

Sivasish Sindiri, http://orcid.org/0000-0003-2516-969X
Qin Tang, http://orcid.org/0000-0002-9487-570X
David M Langenau, http://orcid.org/0000-0001-6664-8318

## Ethics

Animal experimentation: Studies were approved by the Massachusetts General Hospital Subcommittee on Research Animal Care under the protocol #2011N000127 (zebrafish) and #2013N000038 (mouse). Biosafety lentiviral work was approved by the Partners IBC under protocol #2013B000039.

## Additional files

### Supplementary files

• Supplementary file 1. Primers and shRNAs used in this work.

• Supplementary file 2. GREAT analysis identifies commonly bound genomic sites between MYF5 and MYOD in human Rh18 and RD cells.

• Supplementary file 3. Common genomic regions bound by MYF5 and MYOD in ERMS that comprise the 'cyclin-dependent protein kinase holoenzyme complex' module.

• Supplementary file 4. Common genomic regions bound by MYF5 and MYOD in ERMS that comprise the genes found in the 'embryonic skeletal system development' module.

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
