## [Decision Letter]

Thank you for submitting your article "Myogenic Regulatory Transcription Factors Induce Self-renewal and Growth in Rhabdomyosarcoma" for consideration by *eLife*. Your article has been favorably evaluated by Marianne Bronner (Senior editor), Chi Dang (Reviewing editor) and two reviewers.

The reviewers have discussed the reviews with one another and the Reviewing Editor, Chi Dang, has drafted this decision to help you prepare a revised submission.

Summary:

The paper by Tenete et al. reports on a role for Myf5 and MyoD in the self renewal of the cancer stem cell that generate rhabdomyosarcoma (ERMS). Using a combination of zebrafish models, human primary tumour samples and syngeneic mouse engraftment models the authors implicate the myogenic regulators in the propagation of tumours. The authors focused on Myf5 in ERMS initiation and propagation. Using a *kRAS^G12D^* driven in vivo zebrafish ERMS model, the authors demonstrated that enforced expression of *myf5* in *mylpfa*+ cells cooperated with *kRAS^G12D^* to induce differentiated ERMS, with earlier tumor onset and higher tumor penetrance comparing to k*RAS^G12D^*-only undifferentiated ERMS. These differentiated ERMS cells are transplantable, confirming full-transformation of the cells. Interestingly, the ectopic expression of *mylpfa:myf5* transformed differentiated (*myf5*+, *mylpfa*+) ERMS cells into self-renewable tumor propagating cells (TPCs), whereas only the undifferentiated (*myf5*+, *mylpfa*-) cells were TPCs in the *kRAS^G12D^*-only tumors. They then studied Myf5 and MyoD in RMS tumor cell growth using human RMS tumor cell lines (for MYF5 or MYOD) and mouse xenografts (for Myf5). The authors showed that either Myf5 or MyoD, but not both, are expressed in human tumors and cell lines. Knocking-down of MYF5 or MYOD in cell lines reduced cell growth and proliferation. MYF5 knockdown also suppressed growth of mouse xenografts. The authors performed ChIP-seq and showed a few examples of commonly bound target genes of MYF5 and MYOD.

Essential revisions:

1) The implication of self-renewal seems a bit of a stretch. We would like to see a better structured argument that this is the case. Is not most of the data also consistent with myogenic regulators being involved in enhanced tumour propagation, a very interesting fact in its own right?

2) The authors take for granted that we know their system. It is not that intuitive that the *rag2-kRAS* model drives expression in muscle stem cells. While this is of course published, the reader is owed an explanatory sentence or two on this before diving into the main results of the paper.

3) There is confusion about the transgenic approach where *rag2-kRas* and *mylpa-myf5* are co injected into the early embryo leading to the gain of tumour properties described. If these are mosaic injected G0 animals that are analysed then the chances of co-expression of these two plasmids together in the same cell will be very small? How then does it work? Can every tumour in a *rag2-kRas* and *mylpa-myf5* injected animal will be transgenic for both transgenes? This needs more explanation as the experimental approach is not clear and there is not enough detail in the Methods. A discussion of exactly when the *mylpa* promoter is active during muscle differentiation (or a characterisation of it in the context shown) is also missing. Is it really restricted to terminally differentiated cells in the micro-injection mosaic assay used?

4) It would have been better to work out if the upregulation of the myogenic regulators that is evident in transgenic tumours is the exogenously supplied *myf5* or the endogenous gene. We assume that it is not a GFP fusion, but it’s not clear. Can the transgenically supplied *myf5* be tagged to distinguish it? This would also help with point 3.

5) Is there a technical reason why siRNA knockdown and not Crisper CAS9 deletion of MYF5 and MYOD was performed in the human ERMS cells. It would seem if you can transfect siRNA you can make the mutant cells which would be far more convincing.

6) The obvious controls are to knockdown either MYF5 and MYOD in the Human tumour cells not over expressing that specific MRF and see no effect? This would be a great control to guard against non specific effects of RNAi on cell cycle progression (i.e. you just made the cells sick).

Figure 1: *rag2:kRAS^G12^* induced *myf5:GFP*+, *mylpfa*:mCherry+ (yellow) RMS tumor in zebrafish, whereas *rag2:kRAS^G12^* and *mylpfa:myf5* co-expression induced *myf5:GFP*+, *mylpfa*:mCherry- (green) tumor. The loss of *mylpfa*:mCherry suggests that the tumor in panel D is more undifferentiated comparing to the tumor in panel A, which conflicts with panel E and F which indicate that the tumors were more differentiated. The green tumor in panel D also conflicts with the data presented in Figure 2 and Figure 3, that the *myf5*-overexpressing tumors contained >80% red cells. The authors should clarify these points and possible show images that illustrate the findings.

7) Issues with the figures:

Figure 1: The gene expression data in Figure 1 suggested a positive correlation of *myf5, cdh15* and myog expression in zebrafish ERMS. In contrast, Figure 4 (panels A and B, and subsection “MYF5 and MYOD are required for continued tumor growth in human RMS”) clearly showed that high *myf5* expression and high *myod1/cdh15/myog* were separable in human RMS tumors, in that tumors with high *myf5* have relative low levels of *cdh15/myog*. The authors should provide explanation for this difference between the human and zebrafish RMS. What is the expression level of *myod* in these zebrafish ERMS tumors? Are *myod* and *myf5* exclusive in zebrafish *kRAS^G12D^*-tumors?

Figure 2: panels G, H, I and L showed that *mylpfa:myf5*-expressing zebrafish ERMS tumors had significantly reduced proportions of G+ and G+R+ cells compared to *kRAS^G12D^*-only tumors. According to Figure 3, these populations are the TPCs in *mylpfa:myf5*-expressing tumors (G+ and G+R+). Based on the cumulative TPC frequencies in Table 1 and the data in Figure 2, there are much less TPCs in *mylpfa:myf5*-expressing tumors (1/377 in the 0.1% G+ plus 1/639 in the 1.1% G+R+, panel H) than in *kRAS^G12D^*-only tumors (1/146 in the 3.2% G+, panel G). Given panel J and K showing that the *mylpfa:myf5*-expressing tumors arose earlier and grew bigger after transplantation, it suggests that the *mylpfa:myf5*-expressing tumors need less recovery after transplantation or grow faster.

Since Figure 1—figure supplement 2 shows similar fractions of cells in mitosis and similar levels of apoptosis, how do the *kRAS^G12D^* tumors with *mylpfa:myf5*-expression grow faster? In primary *kRAS^G12D^* tumors with or without *mylpfa:myf5*-expression, it would be important to show and compare the growth/proliferation rate in the transplanted secondary tumors. Do the *mylpfa:myf5*-expressing tumors divide faster in transplants? Can the red cells still divide? Do the G+/G+R+ cells show a high level of asymmetric division?

Figure 3: panels K and M, and subsection “Myf5 reprograms differentiated ERMS cells into self-renewing TPCs” showed similar levels of *c-met, cdh15* and *mylpfa* in zebrafish ERMS regardless the *mylpfa:myf5*-expression. This data conflicts with Figure 1 and subsection “Re-expression of *myf5* in zebrafish ERMS cells leads to accelerated tumor onset and increased penetrance”, first paragraph, which showed significantly higher *c-met/cdh15* in the *mylpfa:myf5*-expressing tumors. The authors should explain the difference. The expression of myod should be included in panels K and M.

Figure 4: siRNA and shRNAs were used to knockdown MYF5 and MYOD. Western blots showed that the siRNAs showed slightly better knockdown at protein level. But the EDU labeling and nuclei counts showed that siRNAs resulted in less impaired cell growth. In this case, the authors should perform cDNA rescue experiments to prove the specificity of the siRNA/shRNAs. Also, panel 4K showed si-*MYOD* significantly reduced EDU+ RD cells, but panel L showed that si-*MYOD* did not impair nuclei counts. This point should be discussed, as Figure 4—figure supplement 3 showed that si-*MYOD* didn't affect apoptosis in RD cells.

Figure 4: The authors showed that MYF5 and MYOD are differentially expressed in human tumors and cell lines. How does knocking-down of one affect the expression of the other in the cell lines?

Figure 4—figure supplement 3: "Apoptosis was not increased in si-*MYF5* treated cells (Figure 4—figure supplement 3)". But there is no si-*MYF5* in Figure 4—figure supplement 3 at all. Panel D showed some apoptosis reduction by sh-*MYF* #1 but not by sh-*MYF* #2. Data of sh-*MYF* #3 and si-*MYF* should be shown. Also, panel C showed that si-*MYOD* in Rh18 cells significantly reduced apoptosis, which needs to be explained because i) Rh18 cells express low MYOD, ii) si-*MYOD* did not significantly impair apoptosis in the MYOD-expressing RD cells (panel G). In addition, apoptosis analysis should be provided for 381T, RMS559 and Rh3 cells.

Figure 5: the session is subtitled as 'Myogenic transcription factors are requited for continued xenograft growth'. However, the provided xenograft data was solely about MYF5 in Rh18 cells. How about MYOD? The authors should either provide MYOD data to keep the subtitle, or change the subtitle to 'MYF5 is required". The same applies to the title. The authors should either show the self-renewal data of *myod*, or modify the title.

Figure 6: panel E and Results, last paragraph, suggested that MYOG, MYHC1, CDH15 and CCND2 are commonly bound target genes of MYF5 and MYOD. However, only the binding of MYF5/MYOD to MYOG and CCND2 was shown in panel D. Binding of MYF5/MYOD to MYHC1 and CDH15 should also be shown, either in the same panel or as the figure supplement. Expression levels of MYOD in Rh18 cells and MYF5 in RD cells after si-*SCR* and si-*MYF5* should be included in panel E. In addition, were *myhc1* and *ccnd2* upregulated in *mylpfa:myf5*-expressing zebrafish ERMS tumors (*myog* and *cdh15* were upregulated in Figure 1)?

Figure 6: Results, last paragraph and Discussion, last paragraph suggested that CCND2 is an important common target of MYF5/MYOD. Could CCND2 (partially) rescue the RMS cell growth defect following MYF5/MYOD as shown in Figure 4?

Figure 6 and last sentence of the Abstract. The authors emphasize the shared targets of MYF5 and MYOD. Is there any difference between them, or do they do the same thing? Human tumors express one or the other. Is it the same in *kRAS^G12D^* induced zebrafish tumors? *myod* levels should at least be measured in the zebrafish tumors to tie the zebrafish and human data together. When *myf5* is overexpressed in the zebrafish, inducing earlier onset and faster growth, do the *myod* levels fall?

---

## [Author Response]

*Essential revisions:*

*1) The implication of self-renewal seems a bit of a stretch. We would like to see a better structured argument that this is the case. Is not most of the data also consistent with myogenic regulators being involved in enhanced tumour propagation, a very interesting fact in its own right?*

We have changed the title of the paper and modified the text to better describe our work. When appropriate, we compare sustained tumor growth, which in ERMS is driven by molecularly- defined TPCs, to self-renewal found in normal stem cells. We hope we have struck a better balance in the presentation of our data, with a new focus on defining roles for MYF5/MYOD in sustained tumor growth and propagation as requested.

*2) The authors take for granted that we know their system. It is not that intuitive that the rag2-kRAS model drives expression in muscle stem cells. While this is of course published, the reader is owed an explanatory sentence or two on this before diving into the main results of the paper.*

We have added a new introductory paragraph to better describe the model and how it would be used in manuscript.

*3) There is confusion about the transgenic approach where rag2-kRas and mylpa-myf5 are co injected into the early embryo leading to the gain of tumour properties described. If these are mosaic injected G0 animals that are analysed then the chances of co-expression of these two plasmids together in the same cell will be very small? How then does it work? Can every tumour in a rag2-kRas and mylpa-myf5 injected animal will be transgenic for both transgenes? This needs more explanation as the experimental approach is not clear and there is not enough detail in the Methods.*

We have added a more detailed discussion of the model and approach to the introduction as requested. Briefly, because transgenes integrate into the genome as high copy concatamers, it is possible to deliver up to 3 independent transgenic reporters by microinjection of linearized DNA into one-cell-stage fish. This approach allows co-expression of each transgene in *all* developing tumor cells. We first published this technique in 2008 (Langenau et al., Oncogene. 2008;27(30):4242-8. PMID: 18345029) and provided many rigorous tests to validate our approach in a wide range of cancers, including the *kRAS^G12D^*-driven ERMS model. To date, this microinjection technique has become a well-established approach for the field.

We hope we have done a better job in describing our experimental approach in the revised manuscript and thank the reviewer for pointing out the need to better define our model for the reader.

*A discussion of exactly when the mylpa promoter is active during muscle differentiation (or a characterisation of it in the context shown) is also missing. Is it really restricted to terminally differentiated cells in the micro injection mosaic assay used?*

Others and we have used the *mylpfa*-promoter to drive transgene expression in differentiated muscle cells both in the stable and mosaic transgenic setting (Ignatius et al., 2012; Ju et al., 2003; Storer et al., 2013; Tang et al., 2016). In both settings, the *mylpfa* transgene expression is confined to differentiated ERMS cells. We have added two sentences to the Methods section to highlight this work and hope we have done a better job in pointing the reviewer to the published literature within the main text.

*4) It would have been better to work out if the upregulation of the myogenic regulators that is evident in transgenic tumours is the exogenously supplied myf5 or the endogenous gene. We assume that it is not a GFP fusion, but it’s not clear. Can the transgenically supplied myf5 be tagged to distinguish it? This would also help with point 3.*

To address this reviewer question, we have now completed quantitative real-time PCR gene expression studies in both *kRAS^G12D^*and *kRAS^G12D^+myf5* transgenic tumors, specifically assessing endogenous and total *myf5* expression (which includes transgenically supplied *myf5*).

We find that endogenous *myf5* is elevated 3-fold in *mylpfa:myf5* expressing tumors suggesting a feedback loop can modify expression. By contrast, transgenic *myf5* is highly expressed in *mylpfa:myf5* expressing ERMS as expected (Figure 1).

*5) Is there a technical reason why siRNA knockdown and not Crisper CAS9 deletion of MYF5 and MYOD was performed in the human ERMS cells. It would seem if you can transfect siRNA you can make the mutant cells which would be far more convincing.*

At the time of initiating this work, Crispr/CAS9 approaches were not the norm and to date methods to perform this technique in RMS cells has yet to be reported. Rather our work used at least two shRNAs for each gene and subsequent siRNA knockdown in multiple RMS cell lines. We now provide data showing specificity of knockdown for each factor (Figure 4—figure supplement 4). These approaches are still commonly used in the field. We also point out, that although elegant, Crispr/CAS9 approaches are also subject to off-target effects and require vetting with other approaches.

*6) The obvious controls are to knock down either MYF5 and MYOD in the Human tumour cells not over expressing that specific MRF and see no effect? This would be a great control to guard against non specific effects of RNAi on cell cycle progression (i.e. you just made the cells sick).*

We have completed the requested studies and now show remarkable specificity of our knockdowns. Specifically, we have used siRNA knockdown of MYF5 in ERMS cells that lack its expression and completed comparable experiments using *siMYOD*. We find that siRNA effects are highly specific and do not lead to non-specific cell toxicity (see Figure 4—figure supplement 4). We extended this analysis to a wide array of cell lines and tested each siRNA for functional effects on regulating proliferation, further bolstering the claims of our work.

We did attempt rescue experiments as requested, but had difficulty in simultaneously knocking down gene expression and delivering MYF5 or MYOD.

*Figure 1: rag2:kRAS^G12^ induced myf5:GFP+, mylpfa:mCherry+ (yellow) RMS tumor in zebrafish, whereas rag2:kRAS^G12^ and mylpfa:myf5 co-expression induced myf5:GFP+, mylpfa:mCherry- (green) tumor. The loss of mylpfa:mCherry suggests that the tumor in panel D is more undifferentiated comparing to the tumor in panel A, which conflicts with panel E and F which indicate that the tumors were more differentiated. The green tumor in panel D also conflicts with the data presented in Figure 2 and Figure 3, that the myf5-overexpressing tumors contained >80% red cells. The authors should clarify these points and possible show images that illustrate the findings.*

We apologize for the rendering of fluorescent colors shown in the previous version of this merged image. As rightfully noted, we failed to correctly compensate fluorescence in the image shown in panel 1D. In the revised manuscript, we now provide a full analysis of each fluorescent image panel and a better rendering of the data (See Figure 1, and Figure 1—figure supplement 1).

*7) Issues with the figures:*

*Figure 1: The gene expression data in Figure 1 suggested a positive correlation of myf5, cdh15 and myog expression in zebrafish ERMS. In contrast, Figure 4 (panels A and B, and “MYF5 and MYOD are required for continued tumor growth in human RMS”) clearly showed that high myf5 expression and high myod1/cdh15/myog were separable in human RMS tumors, in that tumors with high myf5 have relative low levels of cdh15/myog. The authors should provide explanation for this difference between the human and zebrafish RMS. What is the expression level of myod in these zebrafish ERMS tumors? Are myod and myf5 exclusive in zebrafish kRAS^G12D^-tumors?*

Although zebrafish provide a powerful tool to uncover human biology, it is not always the case. As correctly noted by the reviewer, zebrafish ERMS that have high transgenic *myf5* transcript expression also express higher levels of all muscle associated genes, including *myoD*. These data likely reflect the artificial model we have developed where *myf5* is re-expressed in differentiated ERMS cells. To date, we have yet to generate a suitable zebrafish model of the human ERMS tumor subtypes that express only *myoD* to directly address the reviewer question.

As requested, we have added *myoD* gene expression data to Figure 1 and Figure 3.

*Figure 2: panels G, H, I and L showed that mylpfa:myf5-expressing zebrafish ERMS tumors had significantly reduced proportions of G+ and G+R+ cells compared to kRAS^G12D^-only tumors. According to Figure 3, these populations are the TPCs in mylpfa:myf5-expressing tumors (G+ and G+R+). Based on the cumulative TPC frequencies in Table 1 and the data in Figure 2, there are much less TPCs in mylpfa:myf5-expressing tumors (1/377 in the 0.1% G+ plus 1/639 in the 1.1% G+R+, panel H) than in kRAS^G12D^-only tumors (1/146 in the 3.2% G+, panel G). Given panel J and K showing that the mylpfa:myf5-expressing tumors arose earlier and grew bigger after transplantation, it suggests that the mylpfa:myf5-expressing tumors need less recovery after transplantation or grow faster.*

*Since Figure 1—figure supplement 2 shows similar fractions of cells in mitosis and similar levels of apoptosis, how do the kRAS^G12D^ tumors with mylpfa:myf5-expression grow faster? In primary kRAS^G12D^ tumors with or without mylpfa:myf5-expression, it would be important to show and compare the growth/proliferation rate in the transplanted secondary tumors. Do the mylpfa:myf5-expressing tumors divide faster in transplants? Can the red cells still divide? Do the G+/G+R+ cells show a high level of asymmetric division?*

Sadly, no living transplant tumors were available to complete the EDU analysis as requested – reflecting a need to generate primary tumors and then perform transplantations, which would require in excess of the three months allowed for resubmission.

However, we were able to complete EDU experiments in primary tumors, uncovering a trend toward higher rates of proliferation in *mylpfa:myf5* expressing ERMS (Figure 1—figure supplement 3). Sadly, these tumors were not generated in the double transgenic fluorescent transgenic background, precluding comment on effects within each specific cell sub-population.

We have also amended this section to better present our results. Our data strongly suggest that the dominant role of exogenously expressed *myf5* is to transform a wider range of differentiated cell types, reflected in the earlier tumor onset and higher penetrance of disease in primary ERMS (Figure 1).

*Figure 3: panels K and M, and subsection “Myf5 reprograms differentiated ERMS cells into self-renewing TPCs” showed similar levels of c-met, cdh15 and mylpfa in zebrafish ERMS regardless the mylpfa:myf5-expression. This data conflicts with Figure 1 and subsection “Re-expression of myf5 in zebrafish ERMS cells leads to accelerated tumor onset and increased penetrance”, first paragraph, which showed significantly higher c-met/cdh15 in the mylpfa:myf5-expressing tumors. The authors should explain the difference. The expression of myod should be included in panels K and M.*

Data shown in Figure 3 denote normalized relative expression between *kRAS^G12D^*alone and *kRAS^G12D^*+ *mylpfa:myf5* expressing tumors. We have used this approach in the past to assess gene expression differences within tumor subpopulations between individual tumors,

reflecting that tumors often have variable levels of total gene expression (Ignatius et al., Cancer Cell2012).

In the example provided, comparisons should only be made to G+ vs. G+R+ cells within a given panel/genotype. For example, these data show that the same molecular markers define immature and mature ERMS cell subfractions in either genotype. By contrast, *myf5* is expressed highly in differentiated cells in panel M as would be expected of *mylpfa:myf5* expressing ERMS. We have provided additional description in the Results and Materials and methods sections to clarify.

We have also included the *myoD* gene expression in these panels as requested.

*Figure 4: siRNA and shRNAs were used to knockdown MYF5 and MYOD. Western blots showed that the siRNAs showed slightly better knockdown at protein level. But the EDU labeling and nuclei counts showed that siRNAs resulted in less impaired cell growth. In this case, the authors should perform cDNA rescue experiments to prove the specificity of the siRNA/shRNAs.*

We have now added additional experiments to assess the specificity of our knockdown. Specifically, we have performed siRNA knockdown of both MYOD and MYF5 in multiple ERMS cell lines. As expected, siRNAs that target expressed genes result in potent impairment of cell cycle, while knockdown of the other non-expressed myogenic factor had no effect on proliferation.

As discussed above, cDNA rescue experiments were attempted but did not lead to conclusive/reproducible results.

*Also, panel 4K showed si-MYOD significantly reduced EDU+ RD cells, but panel L showed that si-MYOD did not impair nuclei counts. This point should be discussed, as Figure 4—figure supplement 3 showed that si-MYOD didn't affect apoptosis in RD cells.*

We have removed this data from the revised manuscript. These data are in fact confusing. This is because it is possible to have potent cell cycle defects that have not yet manifested in reduced cell numbers (i.e. cell cycle is impaired, but cannot be read out as reduced cell number at the time point analyzed). Moreover, siRNA knockdown is transient, and thus late effects manifested as overall reductions in cell number may not be read out. Rather than confuse this issue, we have opted to report long-term growth effects only in shRNA knock down. These experiments used manual quantification of nuclei counts in vitro(Figure 4—figure supplement 3) and tested effects of knockdown xenograft growth in vivo(Figure 5 and Figure 5—figure supplement 1).

*Figure 4: The authors showed that MYF5 and MYOD are differentially expressed in human tumors and cell lines. How does knocking-down of one affect the expression of the other in the cell lines?*

As outlined above, we have now completed these experiments and show that 1) knockdowns are specific to expressed myogenic factor and 2) gene compensation by the other related factor is not observed following knockdown. This was completed in several ERMS cell lines and is now shown in Figure 4—figure supplement 4.

We thank the reviewers for suggesting this important experiment, which greatly strengthens our conclusions that MYOD and MYF5 regulate cell cycle in ERMS cells.

*Figure 4—figure supplement 3: "Apoptosis was not increased in si-MYF5 treated cells (Figure 4—figure supplement 3)". But there is no si-MYF5 in Figure 4—figure supplement 3 at all. Panel D showed some apoptosis reduction by sh-MYF #1 but not by sh-MYF #2. Data of sh-MYF #3 and si-MYF should be shown. Also, panel C showed that si-MYOD in Rh18 cells significantly reduced apoptosis, which needs to be explained because i) Rh18 cells express low MYOD, ii) si-MYOD did not significantly impair apoptosis in the MYOD-expressing RD cells (panel G). In addition, apoptosis analysis should be provided for 381T, RMS559 and Rh3 cells.*

In the previous submission, we made an error in labeling this panel in Figure 4—figure supplement 3. This was indeed siMYF5 and the data showed that knockdown did not lead to elevated apoptosis at the 72 hour time point analyzed.

To address this important reviewer point, we have now repeated our apoptosis experiments at a latter time point (96 hours). This data is now presented in Figure 4—figure supplement 3 and Figure 4—figure supplement 4. We now report that Rh18, RD, and RMS559 cells have elevated apoptosis that is secondary to cell cycle that manifest as early as 48h post infection. We point out that apoptosis phenotypes are indeed variable. For example, we see no change in apoptosis in 381T or Rh3 cells following siMYOD knockdown at 96 hours (Figure 4—figure supplement 4). We conclude that cell cycle defects are the primary effect of either MYF5 or MYOD knockdown and apoptosis is secondary.

*Figure 4—figure supplement 3, panel D showed some apoptosis reduction by shMYF #1 but not by shMYF#2. Data of shMYF#3 and siMYF5 should be shown.*

In addition, apoptosis analysis should be provided for 381T, RMS559 and Rh3 cells.

We have added an independent repeat of shRNA knockdown for all three MYF5 shRNAs performed in RH18 cells (Figure 4—figure supplement 3). As expected, we find that apoptosis is not greatly elevated following knockdown at the 3-day time point, supporting our interpretation that apoptosis is induced secondary to cell cycle defects.

We have also completed the requested apoptosis experiments for siRNA treated 381T, RMS559 and Rh3 cells. Specifically, cells were analyzed at 4 days post-transfection and showed that apoptosis was not reproducibly affected in RMS cells following MYF5 or MYOD knockdown (Figure 4—figure supplement 4). These data again support our idea that cell cycle is the major effect following myogenic factor disruption, with secondary effects leading to elevated apoptosis over time.

*Figure 5: the session is subtitled as 'Myogenic transcription factors are requited for continued xenograft growth'. However, the provided xenograft data was solely about MYF5 in Rh18 cells. How about MYOD? The authors should either provide MYOD data to keep the subtitle, or change the subtitle to 'MYF5 is required". The same applies to the title. The authors should either show the self-renewal data of myod, or modify the title.*

We have added additional data supporting roles for MYOD in regulating RD cell growth using sphere colony formation assays (see Figure 4—figure supplement 3). Sphere colony forming assays are an in vitrosurrogate for assessing effects on TPC frequency and correlate well with effects seen in vivo(Satheesha et al., 2016; Walter et al., 2011).

We have also opted to complete the additional mouse xenograft studies requested by the reviewer, knocking down MYOD in both RD and RMS559 cells. We asked for a five-week extension to complete the work. We find that knockdown resulted in impaired tumor growth, but with different kinetics as found in RH18 cells following MYF5 knockdown. We reconcile these findings by our inability to completely block MYOD expression in these cells by shRNAs and that likely additional mechanisms could be acquired to drive continued growth in MYOD expressing RMS. Moreover, we also see that cell cycle defects appear to be more robust following MYF5 knockdown than MYOD. This latter finding will be the subject of future studies. In total, our work supports important roles for both MYF5 and MYOD in regulating proliferation and when inhibited leads to reduced xenograft growth in mice.

Finally, we also point out that our RD experiments report effects for only one shRNA. Sadly, we made an error when performing the second shRNA knockdown for this cell line (errantly using shRNA control vector). We did not realize this until late in the revision process and were unable to perform additional knockdowns in the time allotted for review. We have opted to include this data because we think it provides an important continuity with data presented throughout the manuscript and was independently validated in vitro, in sphere assays, and using additional ERMS cell lines where two independent shRNAs were used. Should the reviewers disagree with our decision to include this data, we will work with *eLife* to remove the work from the manuscript.

*Figure 6: panel E and Results, last paragraph, suggested that MYOG, MYHC1, CDH15 and CCND2 are commonly bound target genes of MYF5 and MYOD. However, only the binding of MYF5/MYOD to MYOG and CCND2 was shown in panel D. Binding of MYF5/MYOD to MYHC1 and CDH15 should also be shown, either in the same panel or as the figure supplement. Expression levels of MYOD in Rh18 cells and MYF5 in RD cells after si-SCR and si-MYF5 should be included in panel E. In addition, were myhc1 and ccnd2 upregulated in mylpfa:myf5-expressing zebrafish ERMS tumors (myog and cdh15 were upregulated in Figure 1)?*

We are sorry our previous submission was confusing in presenting this data. MYHC1 was not a target of MYF5 or MYOD binding in human ERMS and rather was included to show that shRNA knockdown potently suppressed expression of muscle differentiation genes. To be clear, MYOG, CDH15, and CCND2 were bound by MYOD and MYF5. These same factors were potently downregulated following knockdown in either RD or RH18 cells. As requested the RNA seq and ChIP-seq tracks for these factors have now been provided (Figure 6 and Figure 6—figure supplement 2). We have opted to remove the MYHC1 data from Figure 6 so as not to confuse the reader.

As requested, we have also now completed the requested analysis of *ccnd2* transcript expression in zebrafish ERMS and find a trend toward higher transcript expression in *mylpfa:myf5* transgene expressing ERMS as expected. This was completed using three independent qPCR primers in triplicate (see Figure 6—figure supplement 3). As previously noted, *myogenin* and *cdh15* were also highly expressed in *mylpfa:myf5* expressing ERMS (see Figure 1).

*Figure 6: Results, last paragraph and Discussion, last paragraph suggested that CCND2 is an important common target of MYF5/MYOD. Could CCND2 (partially) rescue the RMS cell growth defect following MYF5/MYOD as shown in Figure 4?*

We do not know if CCND2 is the only direct target of MYOD and MYF5 that regulates overall proliferation defects. Future studies are planned to perform a large-scale, targeted cDNA overexpression screen to identify the factor(s) that rescue MYF5 and MYOD loss in human ERMS.

*Figure 6 and last sentence of the Abstract. The authors emphasize the shared targets of MYF5 and MYOD. Is there any difference between them, or do they do the same thing? Human tumors express one or the other.*

As pointed out above, MYF5 loss leads to severe growth defects and tumor regressions when implanted into NOD/Scid/Il2gr-null mice. By contrast, MYOD knockdown lead to severe reductions in overall xenograft growth but not overall tumor reductions. We do think there are both commonalities in how MYF5 and MYOD regulate proliferation, but it is clear there will be more to the story for MYOD – with likely compensatory or parallel pathways that can regulate proliferation. Future experiments akin to those outlined in the eleventh response to point 7 will be required to address this interesting question.

*Is it the same in kRAS^G12D^ induced zebrafish tumors? myod levels should at least be measured in the zebrafish tumors to tie the zebrafish and human data together. When myf5 is overexpressed in the zebrafish, inducing earlier onset and faster growth, do the myod levels fall?*

We have included the *myoD* qRT-PCR gene expression to Figure 1 and Figure 3 as requested. As discussed above (see first comment to point 7), zebrafish express both *myf5* and *myoD* at the transcription level, differing from what is seen in human RMS. Future studies are planned to see if protein expression differs for these factors in zebrafish ERMS.